# Influence Learning in Complex Systems

**Elena Congeduti**                                    *e.congeduti@tudelft.nl*
*Department of Computer Science*
*Delft University of Technology*
*the Netherlands*

**Roberto Rocchetta**                                    *roberto.rocchetta@supsi.ch*
*Intelligent Energy System Group*
*University of Applied Sciences and Arts of Southern Switzerland*
*Switzerland*

**Frans A. Oliehoek**                                    *f.a.oliehoek@tudelft.nl*
*Department of Computer Science*
*Delft University of Technology*
*the Netherlands*

**Reviewed on OpenReview:** *https://openreview.net/forum?id=tUnyInYbjK*

## Abstract

High sample complexity hampers the successful application of reinforcement learning methods, especially in real-world problems where simulating complex dynamics is computationally demanding. *Influence-based abstraction (IBA)* was proposed to mitigate this issue by breaking down the *global* model of large-scale distributed systems, such as traffic control problems, into small *local* sub-models. Each local model includes only a few state variables and a representation of the *influence* exerted by the external portion of the system. This approach allows converting a complex simulator into local lightweight simulators, enabling more effective applications of planning and reinforcement learning methods. However, the effectiveness of IBA critically depends on the ability to accurately approximate the influence of each local model. While there are a few examples showing promising results in benchmark problems, the question of whether this approach is feasible in more practical scenarios remains open. In this work, we take steps towards addressing this question by conducting an extensive empirical study of learning models for influence approximations in various realistic domains, and evaluating how these models generalize over long horizons. We find that learning the influence is often a manageable learning task, even for complex and large systems. Additionally, we demonstrate the efficacy of the approximation models for long-horizon problems. By using short trajectories, we can learn accurate influence approximations for much longer horizons.

## 1 Introduction

Controlling large distributed systems is a key challenge in artificial intelligence with the potential to impact many fields of application, including computing and information technology (Coulouris et al., 2001), energy systems (Järventausta et al., 2010; Nweye et al., 2023) and transportation (Dimitrakopoulos & Demestichas, 2010). Reinforcement learning (RL) could be promising for such problems as it provides a framework for studying how agents learn and plan under uncertainty in sequential decision making problems. Despite recent successes for large sequential decision-making problems (Kaelbling et al., 1998; Sutton & Barto, 2018; Mnih et al., 2015; Silver et al., 2016), RL techniques still suffer from high sample complexity (Kakade, 2003; Botvinick et al., 2019), which means that agents typically require many trajectories sampled from a simulator to achieve optimal performance. This sample inefficiency becomes a significant hurdle in real-world

scenarios characterized by large, structured environments with complex dynamics. In these contexts, running computationally expensive simulators to collect a sufficient sample of trajectories can be unfeasible.

To mitigate this, influence-based abstraction (IBA) (Oliehoek et al., 2012) offers a principled framework for decomposing the global model of a large factored multiagent problem into small local models, which support lightweight simulations. This approach has proven to be a powerful tool for accelerating online planning (He et al., 2020; 2022), deep RL (Suau et al., 2022b;c), and deep multiagent RL (Suau et al., 2022a). The core idea is to leverage the factored structure of the environment to build a local model for each single agent, assuming that the other agents' policies are fixed. In this way, each agent has a local best response model, which it can use to compute a best response, in terms of local rewards, to the other agents. [1] By abstracting away a large portion of the environment, only a few state variable, referred to as the *local factors*, are retained in each local model. To overcome information loss due to the abstraction process, each local model is complemented with a representation of the *influence*, capturing the effects of external factors and policies of other agents on the dynamics of the local factors.

This abstraction approach has the potential to substantially reduce the state space of the best response problem without any loss in value. In other words, the local agent achieves the same performance when looking for solutions in the smaller local model as in the global model (Oliehoek et al., 2021). However, when part of the system is abstracted away, the local factors no longer preserve the Markov property. To restore Markov transitions, the state of the local model needs to be augmented with the history of appropriate state and action variables. Unfortunately as the number of local histories grows exponentially with time, computing the influence (a conditional marginal inference problem) becomes unfeasible, even for small systems or short-horizon problems. This motivates the idea of employing machine learning methods, which have shown excellent results in sequence modeling (Goodfellow et al., 2016; Gamboa, 2017; Ismail Fawaz et al., 2019), to learn approximate representations of the influence. Intuitively, the more accurate the influence learning models are, the smaller the value loss. Congeduti et al. (2021) prove formally that the value loss is bounded by the approximation error. Therefore, obtaining accurate approximations is crucial for the effectiveness of the entire approach.

Even though influence approximation has been successfully applied in a variety of benchmarks, showing improvements in terms of planning and RL performance (He et al., 2022; Suau et al., 2022b), there exists no systematic analysis of how difficult the influence learning task is in complex scenarios and which learning methods are most effective for this purpose. In addition, maintaining accurate approximations might become particularly difficult for long-horizon tasks. In fact, methods for sequence modeling often struggle with capturing long-term dependencies (Pascanu et al., 2013; Gu & Dao, 2024). Even when advanced techniques prove effective, they might not be computationally feasible, as training could require running costly long simulations. To address this gap and understand for which problems IBA can be successfully applied, we investigate the learning of accurate influence models across various realistic scenarios. We examine which learning aspects are crucial for this purpose. Thus, our aim is not to propose a novel algorithmic approach. Instead, we seek to empirically evaluate existing learning methods for approximating the influence in scenarios that may present challenges, such as complex dynamics and long-horizon problems. As key contributions of our work we show that:

- Approximating the influence might typically be an easy learning task, even for complex and large systems. In fact, small recurrent and temporal convolutional neural networks can learn accurate influence approximations across the investigated benchmarks.

- Learning models can be trained on short-horizon sequences and then deployed to approximate the influence over much longer horizons while maintaining high accuracy. Additionally, we propose a method to estimate the accuracy of the models over long horizons using the short training sequences.

---

[1] Note that such a collection of local problems is subject to the prize of anarchy. That is, it is not guaranteed to lead to optimal system level behavior in terms of total rewards. Nevertheless, it may allow us to deal with a complex optimization system in a tractable way, which is one of the core motivations that sparked the interest in multiagent systems (Shoham & Leyton-Brown, 2008).

In this way, we give positive evidence that influence-based abstraction might provide a feasible approach to dealing with decision making in certain real-life complex systems.

## 2 Background

Here we give a concise introduction to decision making problems formalized as partially observable Markov decision processes (POMDPs), as well as influence-based abstraction.

### 2.1 Factored POMDPs and best response problems

Formally, a POMDP (Kaelbling et al., 1998) is a tuple $\mathcal{M} = (\mathcal{S}, \mathcal{A}, \mathcal{T}, \mathcal{R}, \Omega, \mathcal{O}, b^0, h)$, where $\mathcal{S}$ is the finite state space of the environment, $\mathcal{A}$ is the finite space of available actions, $\Omega$ the observation space, and $h$ the horizon of the problem. Note that, assuming finite state and action space, we restrict our work to discrete state and action variables. Initially, a state $s^0 \in \mathcal{S}$ is drawn from the initial distribution $s^0 \sim b^0$ [2]. At any discrete time step $t \geq 0$, the agent chooses an action $a^t \in \mathcal{A}$, and the state changes according to the distribution $s^{t+1} \sim \mathcal{T}(\cdot \,|\, s^t, a^t)$. The agent then receives a reward modeled as $r^t = \mathcal{R}(s^t, a^t)$ and an observation $o^{t+1} \sim \mathcal{O}(\cdot \,|\, s^{t+1}, a^t)$. A policy $\pi$ encodes the agent's behaviour, mapping the action-observation histories $h^t = (a^0, o^1, \ldots, a^{t-1}, o^t)$ into probability distributions over the action space, i.e. $\pi(h^t) \in \Delta(\mathcal{A})$. The goal of the agent is to optimize the expected cumulative reward for employing a policy $\pi$ given the action-observation history $h^t$: $\mathcal{V}^\pi(h^t) = \mathbb{E}\left[\sum_{k=t}^h r^k \,|\, \pi, h^t\right]$. The policy attaining the maximum value $\mathcal{V}^*$ is called optimal policy and is denoted by $\pi^*$. We focus on specific domains where the state space $\mathcal{S}$ can be decomposed into state variables or *factors*, known as factored POMDPs (Hansen & Feng, 2000).

In line with the concept of joint equilibrium-based search for policies (Nair et al., 2003), we adopt the perspective of a protagonist agent $i$ in problems with multiple interacting agents. This approach is commonly used to solve multiagent problems, as many solution methods rely on finding the best response policies for individual agents (Nair et al., 2003; Hansen et al., 2004; Oliehoek et al., 2014). Specifically, by fixing the policies of all other agents $\pi_{-i}$, the best response problem for agent $i$ can be formulated as a factored POMDP where the actions of the other agents $a_{-i}$ are included into the state space as factors. For the rest of the paper, we will omit the subscript $i$ and assume to address a best response problem for a single agent modeled as a factored POMDP and denoted as $\mathcal{M}_{\text{global}} = (\mathcal{S}, \mathcal{A}, \mathcal{T}, \mathcal{R}, \Omega, \mathcal{O}, b^0, h)$.

### 2.2 A traffic example

We will use as a running example a traffic light control problem represented in Figure 1. A protagonist agent manages the traffic light at intersection 1 within a large road network. The goal of the agent is to minimize traffic congestion at the local intersection by leveraging the information from traffic observations from the surrounding area that is delimited by the red box. This traffic example can be represented as a factored POMDP where the state of the system is represented by the variables that measure the traffic levels at different road stretches as, for instance, the variables $s_{\text{n}\downarrow}$, $s_{\text{w}\leftarrow}$, $s_{\text{src}}$ highlighted in Figure 2(a). Factored POMDP can be more compactly represented by exploiting conditional independence between factors. Specifically, the transitions and observation probabilities can be represented by specific forms of Bayesian networks, the two-stage dynamic Bayesian networks (2DBNs) (Boutilier et al., 1999), as illustrated in Figure 2(b).

Achieving optimal control at intersection 1 does not require simulating all the state variables. For instance, the decisions made at intersection 3 affect indirectly the observations of the agent through the car inflow from the north end (blue arrow). By abstracting away those factors that have only indirect effects on the local intersection, a smaller model can be constructed, as depicted in the red box Figure 2(a). In particular, the local factors highlighted in red, $s_{\text{n}\downarrow}$, $s_{\text{w}\leftarrow}$, represent the incoming and outgoing traffic flows from the north and west ends, respectively. The north end inflow denoted by $s_{\text{src}}$ and depicted in blue is a so-called *influence source* (to be defined formally alter) as it influences directly the local traffic measured by $s_{\text{n}\downarrow}$. The

---

[2]To ease the notation, we use small letters to denote both random variables and their realizations. Capital letters denote sets and functions. For instance, $s^0 \sim b^0$ denotes that the random variable representing the state at time 0 is distributed according to $b^0$.

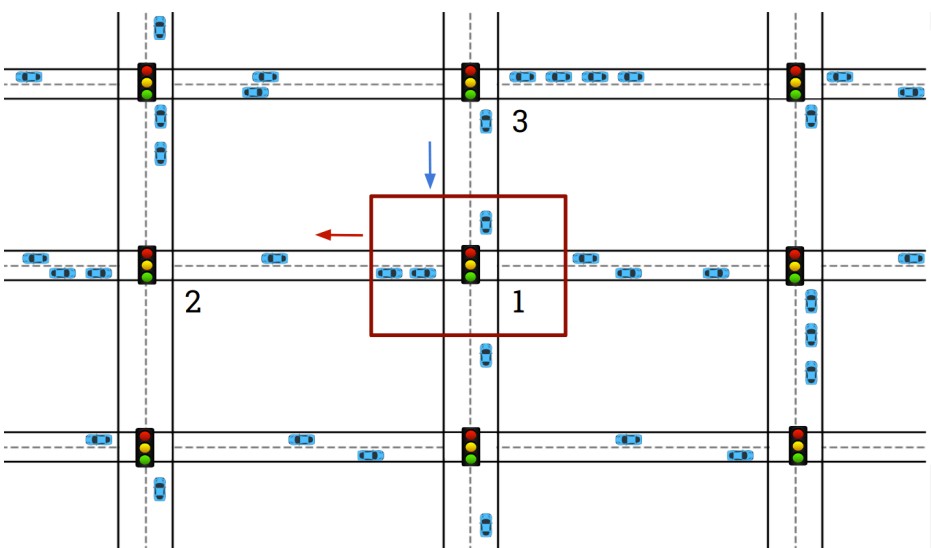

Figure 1: Traffic example. The red square delimits the road segments included in the local model for the traffic light agent 1. The blue arrow represents the effect of the non-local part of the network on the local model and the red arrow the effect of the local traffic on the external part of the system.

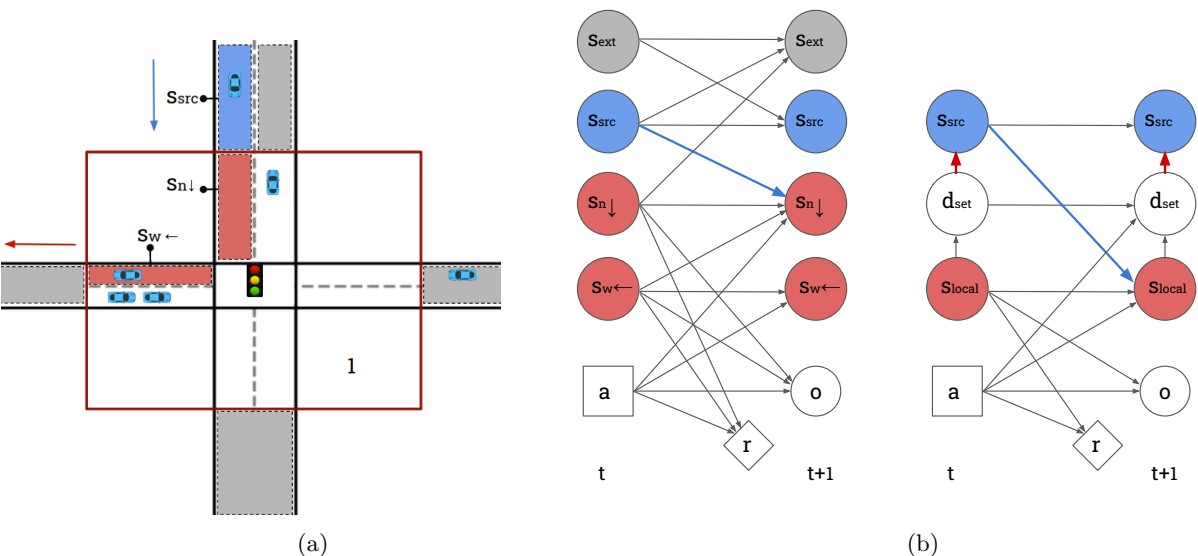

(a)                                                              (b)

Figure 2: Local model for intersection 1 of the traffic network example. (a) graphical representation of the local state variables and dependencies on external factors; (b) 2DBN representations of the local and external state variables (left) and the abstract I-ALM (right).

influence source captures the only information about the external portion of the system necessary to define the local transitions for the north end traffic $s_{n\downarrow}$.

### 2.3 Influence-based abstraction

As we saw in the example, the factorization of the state space allows to leverage the conditional independence between factors. By abstracting away factors that do not directly affect the agent's reward and observations, we can construct a much smaller *local model*. For instance, in the traffic example the state variables that measure the traffic outside of the area delimited by the red box can be discarded. Thus, the state of the environment can be considered as consisting of three components defined as follow.

**Definition 1** *The state $s$ can be decomposed as $s = (s_{ext}, s_{src}, s_{local})$, where $s_{local}$ represents the local factors retained in the model. $s_{src}$ denotes all the external state variables that directly influence the local factors and are called the influence sources. All the remaining factors form an external portion of the state, $s_{ext}$.*

In Figure 2(a), the local factors are depicted in red and the influence source that affects directly the local inflow of cars from the north end is depicted in blue. All the factors that do not affect directly the local traffic are depicted in grey and form $s_{\text{ext}}$. To reduce the complexity of the problem, we want to construct the local model, corresponding to the red area in Figure 2(a), which therefore only contains the local factors. The 2DBN in the left-hand side of Figure 2(b) represents the factorization of the global model state into local factors, influence source and external factors. Note that only the local factors affect the agent's reward and observations. The blue arrow highlight the dependency of the local factor from the external influence sources.

In general the transitions of the local factors depend on the influence sources as $P(s_{\text{loc}}^{t+1}|s_{\text{loc}}^t, s_{\text{src}}^t, a^t)$ which are not part of the local model. For instance, the distribution of $s_{\text{n}\downarrow}$ depends on the north traffic inflow represented by $s_{\text{src}}$. One possible solution is to condition on history. Specifically, we can use the local history to predict the influence source distribution $P(s_{\text{src}}^t|s_{\text{loc}}^0, a^0, \ldots, s_{\text{loc}}^t)$. Interestingly, however, it may not be necessary to condition on the full local state-action history. Oliehoek et al. (2021) show that one can retain the history of only a subset of variables, the *d-set*. The d-set is defined as a set of local variables that d-separates the local factors from the external factors, according to the definition of d-separation for causal graphs in a 2DBN (Koller, 2009; Bishop, 2006). Specifically, the influence sources at time $t$, $s_{\text{src}}^t$, are conditionally independent on the local state and action histories given the d-set, i.e. $P(s_{\text{src}}^t|d_{\text{set}}^t, s_{\text{loc}}^0, a^0, \ldots, s_{\text{loc}}^t) = P(s_{\text{src}}^t|d_{\text{set}}^t)$. Thus, the d-set includes precisely those local factors and actions necessary to predict the influence sources.

In the traffic scenario, the history of the outgoing local traffic measured at the west end $s_{\text{w}\leftarrow}$ affects the future traffic volumes measured by the influence source $s_{\text{src}}$ and thus is part of the d-set. To predict the influence sources, we augment the local state with the local history of west end outflow $s_{\text{w}\leftarrow}$. For the sake of illustration, we will assume that the d-set consists only of those factors, i.e., $d_{\text{set}}^t = (s_{\text{w}\leftarrow}^0, \ldots, s_{\text{w}\leftarrow}^t)$. In the left-hand side of Figure 2(b), the red arrow represents the influence, that is the distribution of the influence source $s_{\text{src}}$ given the $d_{\text{set}}$, $I(s_{\text{src}}^t | d_{\text{set}}^t) \triangleq P(s_{\text{src}}^t | s_{\text{w}\leftarrow}^0, \ldots, s_{\text{w}\leftarrow}^t)$. We can now provide the formal definition.

**Definition 2 (Influence)** *We define the influence $I$ as the conditional probability distribution of the influence sources given the d-set at time $t$, $I(s_{src}^t|d_{set}^t) \triangleq P(s_{src}^t|d_{set}^t)$, defined for any time step $t < h$.*

With these notions, we can define a local model as a POMDP.

**Definition 3 (I-ALM)** *An influence-augmented local model (I-ALM) corresponds to the factored POMDP, $\mathcal{M}_{local} = (\mathcal{S}_{augm}, \mathcal{A}, \mathcal{T}_{local}, \mathcal{R}, \Omega, \mathcal{O}, b^0, h)$, where the augmented state space $\mathcal{S}_{augm}$ comprises the local factors and the d-sets, i.e., $(s_{local}, d_{set}) \in \mathcal{S}_{augm}$. $\mathcal{A}$ is the action space of the agent. The local transitions $\mathcal{T}_{local}$, which model the distributions of the local factors, are defined through the influence $I$ by marginalization over all the possible influence sources $s_{src}$. Precisely,*

$$\mathcal{T}_{local}(s_{local}^{t+1}| s_{local}^t, d_{set}^t, a^t) = \sum_{s_{src}^t} P(s_{local}^{t+1}|s_{local}^t, s_{src}^t, a^t)I(s_{src}^t|d_{set}^t). \tag{1}$$

Note that we assume that the local factors encompass all the variables that directly affect the agent's reward and observations. As a consequence, the reward $\mathcal{R}$ and observation $\mathcal{O}$ are the same functions as those defined for the global model.

Assuming that we can derive the exact influence $I$, the I-ALM $\mathcal{M}_{\text{local}}$ defines a problem equivalent to the global model $\mathcal{M}_{\text{global}}$. That is, the two problems share the same optimal policies. Consequently, solving the smaller I-ALM yields an optimal policy $\pi^*$, which also maximizes the value of the original global problem. We refer to Oliehoek et al. (2021) for a formal proof of this equivalence and an extensive discussion of the IBA concepts introduced in this section. However, deriving the influence for any possible d-set requires solving an exponential number of inference problems in the horizon $h$. For instance, to define the influence for the local traffic model, we would need to compute a number of conditional probability distributions of the order of $(\max s_{\text{w}\leftarrow})^h$, which corresponds to the cardinality of the d-set. This task is practically infeasible even for small values of the variables. This challenge leads to the idea of learning approximations of the influence $\hat{I}$.

## 2.4 Approximate influence-based abstraction

Given a representation $\hat{I}$, an *approximate* influence-augmented local model, a $\hat{I}$-ALM is defined following the same structure of Definition 3 as the factored POMDP $\hat{\mathcal{M}}_{\text{local}} = (\mathcal{S}_{\text{augm}}, \mathcal{A}, \hat{\mathcal{T}}_{\text{local}}, \mathcal{R}, \Omega, \mathcal{O}, b^0, h)$, with local transitions $\hat{\mathcal{T}}_{\text{local}}$ defined by replacing the exact influence $I$ in equation 1 with the approximate influence $\hat{I}$. To construct an $\hat{I}$-IALM, we need to learn $\hat{I}$. For this task, we first simulate $n$ trajectories of influence sources and d-sets from the global simulator to form the training set $\mathcal{D}_h = \left\{ (d_{\text{set}}^0, s_{\text{src}}^0)_i, \ldots, (d_{\text{set}}^h, s_{\text{src}}^h)_i \right\}_{i=1,\ldots,n}$. Now, we define the learning task.

**Definition 4 (Influence learning task)** *Given a training set of influence sources and d-sets trajectories $\mathcal{D}_h$, the influence learning task consists of learning a predictor for the conditional distribution of the influence sources $s_{src}^t$ given the local history in the d-set $d_{set}^t$, $\hat{I}(s_{src}^t | d_{set}^t)$ for any $t = 0, \ldots, h-1$.*

A probabilistic model parameterized by $\theta$, for example a neural network, can be used for this sequence modelling task to approximate the influence as $\hat{I}(\theta) = \hat{I}(s_{\text{src}}^t \mid d_{\text{set}}^t; \theta)$. The learning model is trained to solve the optimization problem formulated as

$$\theta^\star = \arg\min_\theta \mathbb{E}_{\mathcal{D}_h} \left[ \text{CE}(I(\cdot \mid d_{\text{set}}^t) || \hat{I}(\cdot \mid d_{\text{set}}^t; \theta)) \right], \tag{2}$$

where the expectation corresponds to the average cross entropy loss between the target influence and the approximation model.

As mentioned in the introduction, the advantage of this approach lies in the possibility of using a small sample $\mathcal{D}$ from the global simulator to build the local lightweight simulator $\hat{I}$-ALM, which enables more efficient sampling and accelerate the solutions search. Moreover, Congeduti et al. (2021) provide additional support for the approximate abstraction by demonstrating that the value loss for solving the sequential decision-making problem defined by the $\hat{I}$-ALM is bounded by the worst-case KL divergence error of the influence predictions over all the possible d-sets. This bound guarantees that any influence $\hat{I}$ that minimizes the mean cross entropy loss according to equation 2–and thus the mean KL divergence–is aligned with the objective of minimizing the value loss.

## 2.5 State decomposition

In our work, we assume that the decomposition into local and external factors is an engineering choice and thus given, with the constraint that the local factors include all variables that directly affect the reward and the observations. This ensures that reward and observations remain well-defined at local level (Oliehoek et al., 2021). Ideally, the local model should be as small as possible to enable simulation speedups. However, in some cases a larger local model can ease the problem of predicting the influence sources. This may be viewed as a potential limitation, since finding a good trade-off between the information included in the local model and the model size may not be trivial. However, in practice, we have generally found it relatively straightforward to identify sets of local variables that yield good performance (Suau et al., 2022a).

Although we recognize that different decompositions may lead to different outcomes, any choice determines a set of influence sources and d-set, and thus defines an influence learning problem. Assessing the impact of

different decomposition on the complexity of the resulting influence learning problem is quite a challenging task. Therefore, a comprehensive analysis of the effect and efficiency of different choices is beyond the scope of this paper. On the other hand, given a local model, identifying an optimal d-set is relatively straightforward: we select a minimal d-set, as this is also the best choice for the set of input features to the learning models. Using a larger (still d-separating) set of local variables will not improve the accuracy of the learning models. Instead, it would increase the dimensionality of the input of the learning problem, generally making learning more difficult.

## 3 Empirical study of influence learning

The influence learning problem is a sequence prediction task that corresponds to modeling the temporal dependencies between the local and the external factors governed by an underlying 2DBN. This structure differs substantially from the benchmark domains typically used to test sequence modeling methods, such as speech recognition, machine translation, audio classification, music generation, image-video caption generation (Bai et al., 2018; Keneshloo et al., 2019). Thus, this learning problem warrants separate investigation. Based on the intuition that humans are generally quite successful at reasoning about parts of complex systems in isolation, we hypothesize that the local structure of the problems may make the task of learning the influence quite feasible and manageable, even with relatively low-complexity methods for sequence predictions.

To make an impact on real problems with IBA, we need to address two important aspects. First, real-world tasks are often large and complex, which means that we need to explore if approximating the influence is feasible for such problems. Additionally, many real-world tasks often involve systems, such as traffic or networks, that operate continuously, which means that the learning models should work well for very long (or infinite) horizons. For the entire IBA approach to be effective, we need to have sufficiently accurate influence approximations in these situations. Therefore our investigation focuses on two key aspects:

1. Testing the hypothesis that even for large and complex systems, the learning problem may turn into a manageable task that does not require large or complex neural networks.

2. Investigating to what extent the representations learned from trajectories with a short training horizon $h_{\text{train}}$ can approximate well the influence over a much longer deployment horizon $h_{\text{deploy}} \gg h_{\text{train}}$.

### 3.1 Simulation domains

Our experimental setup includes a range of realistic simulation domains, covering diverse situations in terms of number of influence sources to predict, level of 'uncertainty' of their distributions, strength of the dependency between local variables and influence sources, dependency over past time steps, and problem horizons. This variety leads to different characteristics of the learning tasks. Our aim is to determine if this has an impact on the performance of the learning models. We consider four simulation domains designed to test the efficacy of local influence proxies on realistic tasks:

- **Microgrid (MG)**: A realistic application of power system management in the energy engineering sector. Similar to Vázquez-Canteli et al. (2020), in this environment a hundred prosumers (units that both produce and consume electrical power) interact by trading energy to minimize the costs of energy bought from an external grid while meeting the internal power demand. This use case closely emulates problems of practical importance in power engineering, particularly in modern contexts involving distributed energy prosumers, strategies for managing energy storage systems, and recent efforts toward achieving net-zero, positive-energy districts.

- **Traffic grid (TG)**: A traffic light control problem that models the interactions between cars and traffic lights in a road network, as briefly introduced in Section 2.2. This scenario features many external factors which exert a direct influence on the local model. Hence, the influence learning problem corresponds to a high-dimensional prediction task. This transportation network use case

serves as a surrogate for real-world challenges, such as optimizing traffic light operations to minimize queue times and reduce congestion.

- **System admin (SA)**: A multiagent version of the system administrator domain described in Poupart & Boutilier (2004). This environment reproduces a realistic challenge in the information technology domain: a team of system administrators has to manage a network of potentially faulty machines by deciding which machine to reboot.

- **Grab a chair (GC)**: A simple gaming environment introduced by He et al. (2020), where a group of agents competes to obtain one of the available chairs. This environment is a simplified version of the system admin serves as a control scenario for proof-of-concept experiments and preliminary investigations.

We refer the interested reader to Appendix B for a detailed description and visualization of the simulation domains.

## 3.2 Comparison of learning models

We evaluate the performance of different classes of learning models with various degree of complexity (Rocchetta et al., 2023), measured as the network size or number of learnable parameters on the influence learning task.

**Models and benchmark domains used for this experiment.** Many successful sequence prediction methods for sequential predictions are based on neural network with recurrent and temporal convolution components (Karim et al., 2017; Bai et al., 2018; Ismail Fawaz et al., 2019). We evaluate several classes of models including recurrent models such as long short-term memory (LSTM) (Hochreiter & Schmidhuber, 1997) and gated recurrent unit (GRU)(Cho et al., 2014), as well as temporal convolution-based models like temporal convolutional network (TCN) (Lea et al., 2017) and fully convolutional network (FullyConv) (Long et al., 2015). To compare these models with a simpler non-recurrent baseline, we assess a one-layer fully connected linear network, a logistic regression (LogReg) model. We exclude more complex and larger models, such as transformers (Vaswani et al., 2017; Wen et al., 2022), as we assume, and demonstrate, that small and simple models can provide accurate influence representations. For each model class, we compare different sizes, ranging from lightweight networks with less than 100 parameters scaling up to the size a one-layer linear neural network counting more than one million parameters. The hyperparameters defining the network architectures have been chosen to ensure a fair comparison between learning models with the same network size. Since our goal is to investigate whether relatively small and simple models can achieve satisfactory performance, we focus on models with a small number of layers. This choice is also motivated by the fact that deeper networks typically require a larger amount of global simulator trajectories to be trained effectively (Bengio et al., 2013). Moreover, the choice of the network depth depends on the model class. Specifically, convolutional-based models are known to benefit from deeper architectures to achieve better performance and to ensure full receptive field coverage (LeCun et al., 1998; He et al., 2016). For the details of the architectures and sizes used in the different domains, we refer to Table 5, 6, and 7 in Appendix D. The models are evaluated over three domains: microgrid, system admin, and traffic grid. For each domain, we choose the scenario features, including the number of interacting agents $N$, the protagonist agent $i$, the policies of the external agents $\pi_{-i}$, the problem horizon $h$, the initial distribution $b^0$ and other domain-specific parameters that define the transition and reward functions. These scenarios are designed to cover a range of diverse characteristics in terms of problem size, time length, prediction dimension and stochasticity for the influence learning task. Specifically, the microgrid represents a large scenario where 100 agents interact within a power grid; the traffic grid scenario is a real-world traffic control application that turns into a high dimensional learning task; and the influence learning problem for the system admin simulator is a long-horizon forecasting problem.

**Experimental setup.** We use a random exploratory policy for the local agent $\pi_i^{\mathrm{Exp}}$ which affects only the data distribution of the training set. We collect $n = 500$ trajectories of influence sources and d-sets from the global simulator to form the training set $\mathcal{D}_h$. The different models are trained to approximate the influence

as $\hat{I}$, using the mean cross entropy as the training loss. We employ $h$ independent logistic regression models, each one representing the influence $\hat{I}(s_{\mathrm{src}}^t | d_{\mathrm{set}}^t)$ for a specific time step $t \in \{0, \ldots, h-1\}$. We adopt standard optimization techniques, including the ADAM optimization algorithm, linear decay of the learning rate and grid search over the space of initial and final learning rates. For each scenario, a fixed number of epochs and the batch size are selected. Details on the scenarios and hyperparameter configurations are provided in Table 3 and Table 4 in Appendix D, respectively.

**Performance metrics.** The model performance is assessed on an a test set consisting of $n_{\mathrm{test}}$ independent global model trajectories. We use the cross entropy test error, which is computed as the mean over time and sum over the influence sources of the cross entropy estimators. Specifically, for each factor $s_{\mathrm{src},j}^t$ of the influence sources $s_{\mathrm{src}}^t = (s_{\mathrm{src},1}^t, \ldots, s_{\mathrm{src},J}^t)$, an estimator of the cross entropy error $\mathrm{CE}(I(s_{\mathrm{src},j}^t | d_{\mathrm{set}}^t), \hat{I}(s_{\mathrm{src},j}^t | d_{\mathrm{set}}^t))$ at time $t$ is computed over the test sample. Then, the sum over the $J$ influence sources provides the error for a given time step $t$ according to the following formula:

$$\mathrm{CE}^t(I, \hat{I}) = \sum_{j=1}^{J} \left[ -\frac{1}{n_{\mathrm{test}}} \sum_{i=1}^{n_{\mathrm{test}}} \ln \hat{I}(s_{\mathrm{src},j,i}^t | d_{\mathrm{set},i}^t) \right]. \tag{3}$$

With a slight abuse of notation, we denote both the probability distribution of individual influence sources as $\hat{I}(s_{\mathrm{src},j}^t | d_{\mathrm{set}}^t)$ and the joint distribution as $\hat{I}(s_{src}^t | d_{\mathrm{set}}^t)$. The cross entropy error is obtained by averaging the errors defined per time step in equation 3 as follows:

$$\mathrm{e}(h) = \frac{1}{h} \sum_{t=0}^{h-1} \mathrm{CE}^t(I, \hat{I}). \tag{4}$$

We also measure the wall-clock training times (WCTTs) to assess which model provides the best trade-off between accuracy and training time.

### 3.3 Generalization beyond the training horizon

In many real-world applications, an approximate model $\hat{I}$ should perform well over long horizons. Examples of problems that require long-term simulations include long horizon planning (Simeonov et al., 2020; Pertsch et al., 2020), episodic reinforcement learning (Dann & Brunskill, 2015) and tasks characterized by sparse reward signals (Riedmiller et al., 2018). Although it might be feasible to gather a sufficient number of trajectories from the global simulator to approximate well the influence, this may no longer be the case when dealing with long horizon problems as long-term dependencies add further complexity to the learning task.

Our approach to address this issue is to leverage learning models to generalize the influence representations beyond the training horizon. This intuition is grounded in the ergodic theory for MDPs (Puterman, 2014; Morton & Wecker, 1977; Kearns & Singh, 2002). Under ergodic assumptions, the Markov dynamics induced by a joint policy $\pi = (\pi_1, \ldots, \pi_N)$ converge to a unique stationary distribution independently of the initial distribution. The mixing time $t_{\mathrm{mix}}$, refers to the number of time steps required for the state distribution to approximate the stationary distribution within a specified tolerance. Such mixing time is determined solely by intrinsic properties of the system and is independent of the initial conditions. This argument supports the concept of a *stationary influence* as the limit of the conditional distributions of the sequence $\{I(y_{\mathrm{src}}^t | d_{\mathrm{set}}^t)\}_t$. In other words, when the system has sufficiently mixed, the influence approaches a time-independent function, representing the distribution of the influence sources at equilibrium. In this scenario, global trajectories with horizon $h_{\mathrm{train}} > t_{\mathrm{mix}}$ contain sufficient information to learn the stationary influence. Therefore, if the mixing time is sufficiently short, learning models can capture the stationary influence using short training sequences and use it to predict effectively the influence over (indefinitely) longer deployment horizons $h_{\mathrm{deploy}} \gg h_{\mathrm{train}}$.

How to choose an appropriate training horizon for this task remains an open question. We expect predictions to improve as the training horizon $h_{\mathrm{train}}$ approaches the deployment horizon $h_{\mathrm{deploy}}$. However, we aim to limit the computational effort for running long global simulations. Therefore, the training horizon must offer a good trade-off between the quality of approximations and the length of the training sequences. Specifically,

we seek an optimal training horizon $h^*_{\text{train}}$ as the minimum number of time steps required to guarantee accurate predictions across the entire deployment horizon. Intuitively, the optimal horizon $h^*_{\text{train}}$ depends on the system's mixing properties and the ability of the learning models to capture the stationary distribution. Specifically, we verify that it corresponds to the sum of the mixing time $t_{\text{mix}}$ and a few additional time steps needed for the learning models to generalize the experience encountered after mixing.

**Models and benchmark domains used for this experiment.** LSTM and FullyConv networks are employed for the experiments, as they achieve the best performance in the experiments of the previous section on the influence learning tasks. The architectural hyperparameters are selected to ensure that the learning models share the same size. Tables 9 and 10 in Appendix D report the optimization and network architecture details, respectively. We use the grab a chair and traffic grid domains as described in Section 3.1. Grab a chair serves as controlled environment where a known stationary influence is reached after a few time steps of the mixing time. Specifically, we consider two different scenarios: the first with $N = 4$ interacting agents denoted as GC4, and the second with $N = 11$ agents denoted as GC11. We assume that the agents are ordered by index and each external agent $2, 3 \ldots$ copies the action of the preceding agent at previous time step (see Figure 12 in Appendix B). For instance, agent 3 at time $t$ copies the last action of agent 2, which is, in turn, the action of the local agent 1 at time $t-2$. In general, for any agent $i$, $a^t_i = a^{t-i+1}_1$. Note that the external agent's policies are arbitrary features of the experimental domains and, as such, can be chosen to induce various influence learning tasks. This ad-hoc choice ensures that after the mixing time $t_{\text{mix}} = N - 1$ the system converges to a known stationary distribution determined by the local agent policy, i.e. $a^t_i \sim \pi^{\text{Exp}}_1$ for $t \geq N - 1$. As a result, the influence is a time independent deterministic function of the last $N$ local actions determined by the following formula

$$I(a^t_2, a^t_N | a^0_1, \ldots, a^{t-1}_1) = \mathbb{P}(a^t_2, a^t_N | a^{t-1}_1, a^{t-N+1}_1) = \delta_{a^{t-1}_1}(a^t_2)\delta_{a^{t-N+1}_1}(a^t_N) \qquad \text{for } t \geq N - 1. \qquad (5)$$

where $\delta_a$ denotes the Dirac distribution centered on the action $a$. Such explicit expression for the stationary influence allows us to analyze the model learning for the different training horizons. We also use the traffic grid domain to validate our arguments in a more realistic environment where no prior knowledge of stationarity and mixing time is available. Details on the scenarios, training and deployment horizons can be found in Table 8 in Appendix D.

**Experimental setup.** To empirically validate these arguments, we proceed as follows. We consider a set of $K$ training horizons $H = \{h_1, \ldots, h_K\}$ with $h_1 \leq \cdots \leq h_K$. We employ a random exploratory policy $\pi^{\text{Exp}}$ for the local agent and collect $n$ global trajectories of d-sets and influence sources with horizon $h_K$. Then, for each model class, we train $K$ learning models, such that the $k$-th model uses the training set $\mathcal{D}_{h_k}$, which consists of trajectories up to horizon $h_k$, to learn the approximate influence $\hat{I}_k(\theta)$. To assess the generalization ability, we test the models over longer trajectories. For this purpose, we collect an independent test sample of trajectories with horizon $h_{\text{deploy}} \gg h_K$ denoted by $\mathcal{D}_{h_{\text{deploy}}}$.

**Deployment, test and test-tail error.** We define the function $e^k(h)$ to represent the generalization error of the $k$-th model when tested over $h$ deployment steps, as given by the following formula:

$$e^k(h) = \frac{1}{h}\sum_{t=0}^{h-1} \text{CE}^t(I, \hat{I}_k(\theta)), \qquad (6)$$

where the cross entropy term for time step t is defined in equation 3. Note that for $h = h_{\text{deploy}}$, the error $e^k(h_{\text{deploy}})$ corresponds to the average error over $h_{\text{deploy}}$ time steps. This error measures how well (on average) the $k$-th model generalizes the influence over the entire deployment horizon. While for $h = h_k$, the error $e^k(h_k)$ represents the average error of the $k$-th model over $h_{\text{train}} = h_k$ time steps. We refer to these errors as the *deployment error* and the *test error*, respectively, and denote them as follows:

$$e^k_{\text{deploy}} := e^k(h_{\text{deploy}}). \qquad\qquad e^k_{\text{test}} := e^k(h_k) \qquad (7)$$

A learning model that shows good long-term performance should maintain the deployment error close to the test error. Additionally, increasing the training horizon should not significantly reduce the deployment error.

We are also interested in investigating how to estimate the deployment error from the short training trajectories. While the test error might intuitively seem a good candidate estimator, it is significantly affected by the error terms before the system has mixed. To illustrate this, we distinguish the cross entropy error terms in equation 6 into two time scales: the short time scale of the mixing time and the long time scale of the deployment horizon. This distinction allows us to break the error function into two terms as follows:

$$\mathrm{e}^k(h) = \frac{1}{h}\left[\sum_{t=0}^{t_{\mathrm{mix}}-1}\mathrm{CE}^t(I,\hat{I}_k(\theta)) + \sum_{t=t_{\mathrm{mix}}}^{h}\mathrm{CE}^t(I,\hat{I}_k(\theta))\right]. \tag{8}$$

The first term accounts for the short time errors over the influence before mixing and the second for long time errors which represent the model's performance over the stationary influence. Given that $h_{\mathrm{deploy}} \gg t_{\mathrm{mix}}$, the deployment error is mainly affected by the second term of equation 8, while the contribution of the short time errors is negligible. Conversely, when the training horizon $h_{\mathrm{train}}$ is close to the mixing time, the first time errors predominantly determine the test error. These errors do not reflect the model's performance on the stationary influence, causing the test error to deviate from the deployment error.

The key idea for deriving a more accurate estimator is to neglect the errors over the initial time steps. However, the appropriate number of time steps to discard depends on the exact mixing time of the system, which is generally unknown and difficult to estimate. Therefore, we consider different increasing values of the training horizon $h_k$ to test the hypothesis that for certain $h_k > t_{\mathrm{mix}}$, the 'tails' of the error represented in equation 8 are close enough to the deployment error. Specifically, we introduce the *test-tail error*, which is computed as the average error over a window of $l$ time steps, according to the following formula:

$$\mathrm{e}_{\mathrm{tail}}^k = \frac{1}{l}\sum_{t=h_k+1}^{h_k+l}\mathrm{CE}(I,\hat{I}_k(\theta)). \tag{9}$$

In other words, for training horizon $h_k$, $l$ additional time steps are collected in the test set $\{((d_{\mathrm{set}}^{h_k+1}, s_{\mathrm{src}}^{h_k+1}), \ldots, (d_{\mathrm{set}}^{h_k+l}, s_{\mathrm{src}}^{h_k+l}))_i\} \subset \mathcal{D}_{h_{\mathrm{deploy}}}$ and used to compute the test-tail error. We empirically show that in our scenarios the error $\mathrm{e}_{\mathrm{tail}}$ offers a good estimate of $\mathrm{e}_{\mathrm{deploy}}$ for $l = 1$, and can thus be used to assess the quality of the model predictions over the deployment horizon. However, we recognize that increasing the number of time steps, i.e. $l > 1$ might prevent disruptive effects of anomalies and give more stability to the test-tail error. Additionally, we show how this error can be used to search for an optimal training horizon $h_{\mathrm{train}}^*$.

## 4 Results and discussion

Here, we present the empirical results for the two sections of the experiments presented.

### 4.1 Comparison of learning models

We start by analyzing the test error of different model classes and sizes as a function of the number of epochs for the three domains. Together with the training curves, we look at the final test errors and wall-clock training times. To compare the model classes, we select one network size for each class from the Pareto optimal solutions (Miettinen, 1999) of the bi-objective problem of minimizing test error and training time. Finally, we look at the progression of the test error over training time to compare the learning speed and accuracy of the model classes.

**Effect of model class and size.** Figure 3 depicts the progression of the test errors, computed according to equation 4 over the training epochs of the four classes of models (rows) in the three problem domains (columns). For each class and benchmark domain the training curves for different network sizes are illustrated as average test error with standard error over ten iterations of the experiment. The final test errors and training times are reported in Table 1 as averages over the ten iteration of the experiment. The corresponding standard errors are reported for the cross entropy errors but not for the wall-clock training times, as they are negligible.

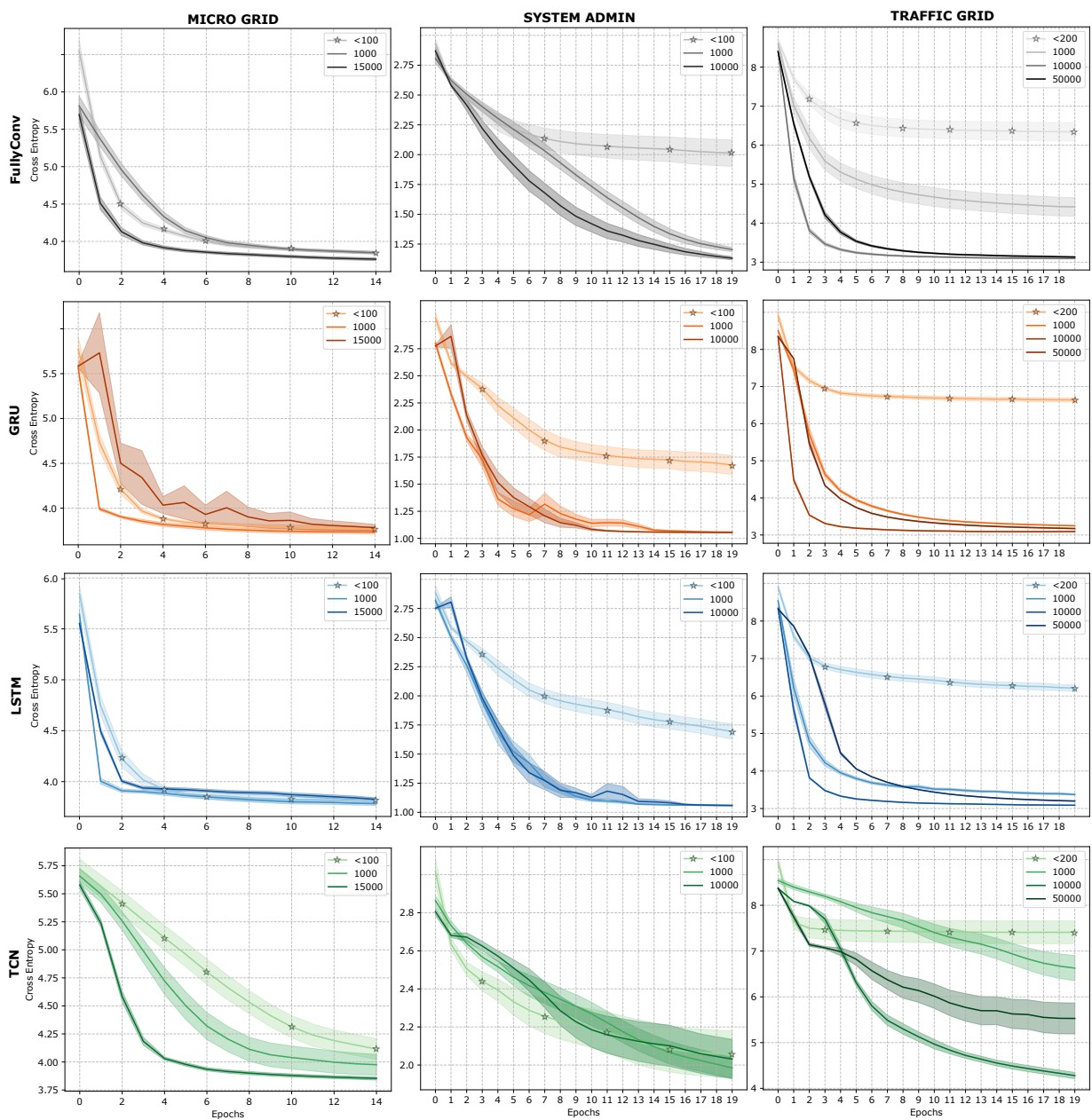

Figure 3: Effect of model size and class on the different benchmark problems. Each panel depicts the cross entropy test error over the training epochs with standard error over ten iterations for different network sizes. The columns represent the three different test environments and the rows the classes of the learning model.

For the microgrid, the training curves and final test errors show no significant advantages from larger model sizes, except in the case of the TCN model class. In fact, the recurrent-based models, such as LSTM and GRU, do not benefit from increased sizes. Conversely, increasing the TCN size from fewer than 100 parameters to 15,000 parameters significantly improves the performance, reducing the test error from 4.11 to 3.85. For the FullyConv model, a bigger size leads to a slight performance improvement but triples the training time from 0.46 to 1.55 seconds. These results align with the observation that temporal convolutional networks typically benefit from deeper architectures and thus require more parameters than recurrent networks (Bai et al., 2018). Especially for long sequences, convolutional networks need several layers to reach a full receptive field, that is to ensure that the entire history-length of the input sequence is processed to make predictions.

Despite the higher number of parameters, the TCN model have typically low training times. In fact, the TCN model with 15,000 parameters achieves accuracy to the other model classes in about the same training time. Figure 3 displays also the results on the system admin and traffic grid domains in the second and third columns, respectively. In contrast with the microgrid domain, very small recurrent (LSTM, GRU) and FullyConv models do not have enough capacity to accomplish the learning tasks. This is due to the higher dimensionality of the space of influence sources and d-sets in the traffic domain and the longer problem horizon for the system admin. These features increase the complexity of the two learning problems compared to the microgrid case.

| | CE | | | WCTT | | |
|---|---|---|---|---|---|---|
| | number of parameters | | | number of parameters | | |
| models | $\leq 100$ | 1000 | 15000 | $\leq 100$ | 1000 | 15000 |
| LSTM | $3.81 \pm 0.03$ | $3.78 \pm 0.02$ | $3.82 \pm 0.03$ | 0.98 | 1.18 | 2.26 |
| GRU | $3.77 \pm 0.03$ | $3.73 \pm 0.03$ | $3.78 \pm 0.03$ | 0.95 | 1.13 | 2.15 |
| TCN | $4.11 \pm 0.03$ | $3.98 \pm 0.03$ | $3.85 \pm 0.04$ | 0.47 | 0.59 | 1.31 |
| FullyConv | $3.85 \pm 0.03$ | $3.85 \pm 0.03$ | $3.76 \pm 0.03$ | 0.46 | 0.65 | 1.55 |
| LogReg | - | - | $3.85 \pm 0.04$ | - | - | 0.95 |

(a) Microgrid

| | CE | | | | | WCTT | | | | |
|---|---|---|---|---|---|---|---|---|---|---|
| | number of parameters | | | | | number of parameters | | | | |
| models | $\leq 200$ | 1K | 10K | 50K | $1M$ | $\leq 200$ | 1K | 10K | 50K | $1M$ |
| LSTM | $6.2 \pm 0.4$ | $3.4 \pm 0.3$ | $3.1 \pm 0.3$ | $3.2 \pm 0.3$ | - | 20.1 | 20.9 | 25.4 | 40.2 | - |
| GRU | $6.6 \pm 0.6$ | $3.3 \pm 0.2$ | $3.1 \pm 0.3$ | $3.2 \pm 0.3$ | - | 21.4 | 22.2 | 27.5 | 40.6 | - |
| TCN | $7.4 \pm 0.2$ | $6.6 \pm 0.0$ | $4.3 \pm 0.4$ | $5.5 \pm 0.5$ | - | 6.0 | 7.4 | 8.2 | 14.3 | - |
| FullyConv | $6.3 \pm 0.2$ | $4.4 \pm 0.3$ | $3.1 \pm 0.3$ | $3.1 \pm 0.3$ | - | 6.5 | 7.9 | 10.3 | 17.0 | - |
| LogReg | - | - | - | - | $5.6 \pm 0.3$ | - | - | - | - | 35.3 |

(b) Traffic grid

| | CE | | | | WCTT | | | |
|---|---|---|---|---|---|---|---|---|
| | number of parameters | | | | number of parameters | | | |
| models | $\leq 100$ | 1K | 10K | 3M | $\leq 100$ | 1K | 10K | 3M |
| LSTM | $1.69 \pm 0.04$ | $1.06 \pm 0.04$ | $1.05 \pm 0.04$ | - | 13.7 | 18.5 | 32.4 | - |
| GRU | $1.68 \pm 0.04$ | $1.06 \pm 0.04$ | $1.05 \pm 0.04$ | - | 13.1 | 18.3 | 33.9 | - |
| TCN | $2.06 \pm 0.05$ | $1.98 \pm 0.03$ | $1.05 \pm 0.04$ | - | 3.6 | 5.1 | 12.7 | - |
| FullyConv | $2.01 \pm 0.04$ | $1.20 \pm 0.05$ | $1.05 \pm 0.04$ | - | 3.9 | 5.2 | 12.3 | - |
| LogReg | - | - | - | $1.97 \pm 0.08$ | - | - | - | 23.2 |

(c) System admin

Table 1: Cross entropy test error with standard error and wall-clock training time (s) for the microgrid, traffic grid and system admin computed over ten iterations.

**Pareto optimal sizes and learning speed.** Figure 4 shows the Pareto frontiers for the model classes (left panels) where each marker represents the test error and training time for a specific size. The dashed lines indicate the Pareto fronts for each class, defined by the sizes that are not strictly dominated by any other in the same class. Among the Pareto optimal solutions, we select and analyze specific sizes which provides a good trade-off between error and training time. These are indicated by the red markers. On the right-hand side of Figure 4, we collect the test error over training time for the selected sizes and compare them with two baseline models: a random model and LogReg.

For the microgrid, the Pareto fronts show that the LSTM, GRU and FullyConv models achieve approximately the same accuracy for the different sizes. Thus, the smaller sizes are selected and highlighted in grey in Table 1, as they also offer the lowest training time. Conversely, the TCN model's error drops significantly with

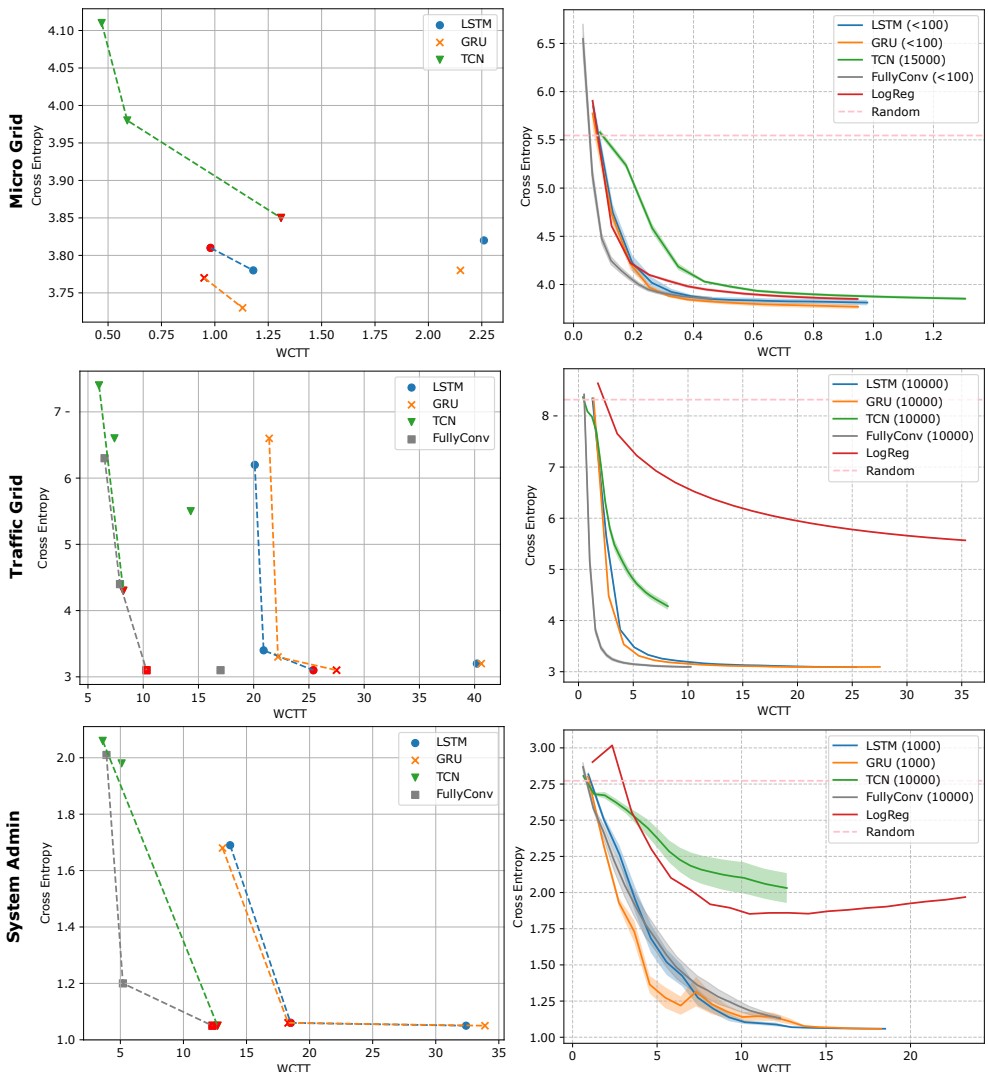

Figure 4: Left panels: Pareto frontiers for the model classes LSTM, GRU, TCN and FullyConv for the cross entropy test error on the y-axis and wall-clock training time on the x-axis for the domains MigroGrid, TrafficGrid and System admin. The sizes selected are represented with red markers and correspond to the cells highlighted in Table 1. Right panels: cross entropy test error over wall-clock training time for the sizes of the network selected for each model class. The dashed line represents the cross entropy error of the baseline random classifier.

larger network size, leading to the choice of size 15,000 for that class. When comparing the training curves of the selected models, we observe similar learning speed and final accuracy across all models. Notably, the logistic regression model achieves comparable performance to the other more complex learning models. One explanation for this result is that, despite the complex nature of the realistic microgrid scenario, the corresponding influence learning task is relatively simple: a low-dimensional forecasting problem with influence sources weakly dependent on the local history. Even scaling up the microgrid to include many more units would still result in a low dimensional and simple influence learning task. This situation is representative of several complex realistic scenarios where the local model is well-decoupled from the rest of the system, with only a few external influence sources weakly affecting the local dynamics.

A different situation arises in the traffic grid and the system admin domains which induce more complex influence learning tasks in terms of dimensionality and problem horizon, respectively. In these domains, the

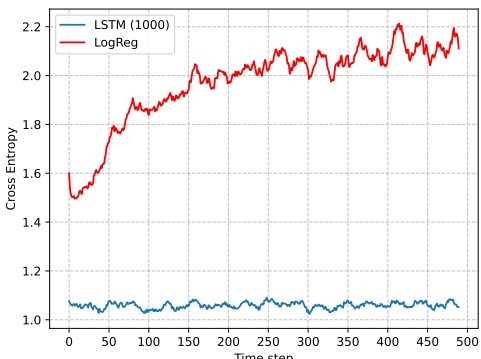

Figure 5: Comparison of cross entropy test error across different time steps in the system administration domain between an LSTM model with 1000 parameters (blue) and the baseline logistic regression model (red). The curves show that the error of the logistic regression increases over time steps, while the LSTM maintains a stable error.

linear regression and TCN models show their limitations. In the right panels of Figure 4), the red training curves show that the logistic regression fails to achieve good accuracy levels. Although the green training curves for the TCN model have not yet reached convergence, the learning is much slower compared to the FullyConv, LSTM and GRU models. However, even for these problems characterized by higher dimension or longer horizons, relatively small architectures for LSTM, GRU and FullyConv models (between 1000 and 10000 parameters) achieve good performance.

**Error over time steps.**  To further analyze the limitation of LogReg model for long horizon problems, we plot in Figure 5 the test error per time step of the logistic regression compared to LSTM for the system admin domain. While the LSTM error remains relatively constant, the error for logistic regression increases over the time steps. This suggests that the logistic regression's ability to represent the influence diminishes with longer horizons. One possible explanation is that linear models struggle to accurately capture increasingly nonlinear relationships between local variables and influence sources as the problem horizon becomes longer.

## 4.2  Generalization beyond the training horizon

First, we look at the deployment errors for specific training horizons to show that it is possible to generalize the influence approximations over long deployment horizons. Then, we analyze the effect of the training horizon on the accuracy of the long-term approximations by looking at the deployment error and training curves for different training horizons. Finally, we compare deployment, test and tail-test error for different training horizons to identify which error serves as a reliable proxy for the deployment error.

Table 2 shows the average deployment errors $e_{\mathrm{deploy}}$ with corresponding standard errors for specific choices of $h_{\mathrm{train}}$, estimated over ten iterations of the entire experiment. The performance of the models in the three scenarios is compared with the error of a random classifier. The choice of a suitable training horizon is

| scenario | learning model | | | | |
|---|---|---|---|---|---|
| | LSTM | FullyConv | Random | $h_{\mathrm{train}}$ | $h_{\mathrm{deploy}}$ |
| GC4 | $0.002 \pm 0.002$ | $0.027 \pm 0.026$ | 1.38 | 6 | 200 |
| GC11 | $0.002 \pm 0.001$ | $0.025 \pm 0.01$ | 1.38 | 22 | 200 |
| TG | $3.68 \pm 0.06$ | $3.98 \pm 0.07$ | 8.32 | 30 | 500 |

Table 2: Mean and standard error of the deployment error over ten iterations.

domain-dependent: in the grab a chair scenario with 4 agents, $h_{\text{train}} = 6$ training steps are sufficient to get deployment error close to 0 while for 11 agents, the models require a longer training horizon $h_{\text{train}} = 22$. Such choices have been driven by the idea that the training horizons need to be longer than the mixing times, which corresponds to $t_{\text{mix}} = 3$ and $t_{\text{mix}} = 10$, respectively. The results show that a training horizon slightly longer than the mixing time ensures cross entropy errors close to zero over a much longer deployment horizon. In other words, few training time steps of experience after mixing are sufficient for the models to generalize the deterministic stationary influence over 200 deployment time steps. For the traffic grid domain, the results are less straightforward to interpret. In fact, the cross entropy error depends on the entropy of the target influence sources distributions, which are unknown. However, for $h_{\text{train}} = 30$ the average errors over 500 deployment time steps in Table 2 are significantly lower than the random classifier error. Also, they are quite close to the errors computed over a much shorter test horizon and reported in Table 1b. This leads to conclude that 30 training steps are sufficient to learn good long-term influence approximations. In summary, for every scenario considered, we found a sufficient number of training steps $h_{\text{train}}$ to learn influence approximations for much longer deployment horizon $h_{\text{deploy}}$ ensuring small deployment error. To gain a concrete and visual understanding of the predictions of the models reported in Table 2, we refer to the videos in Appendix C.

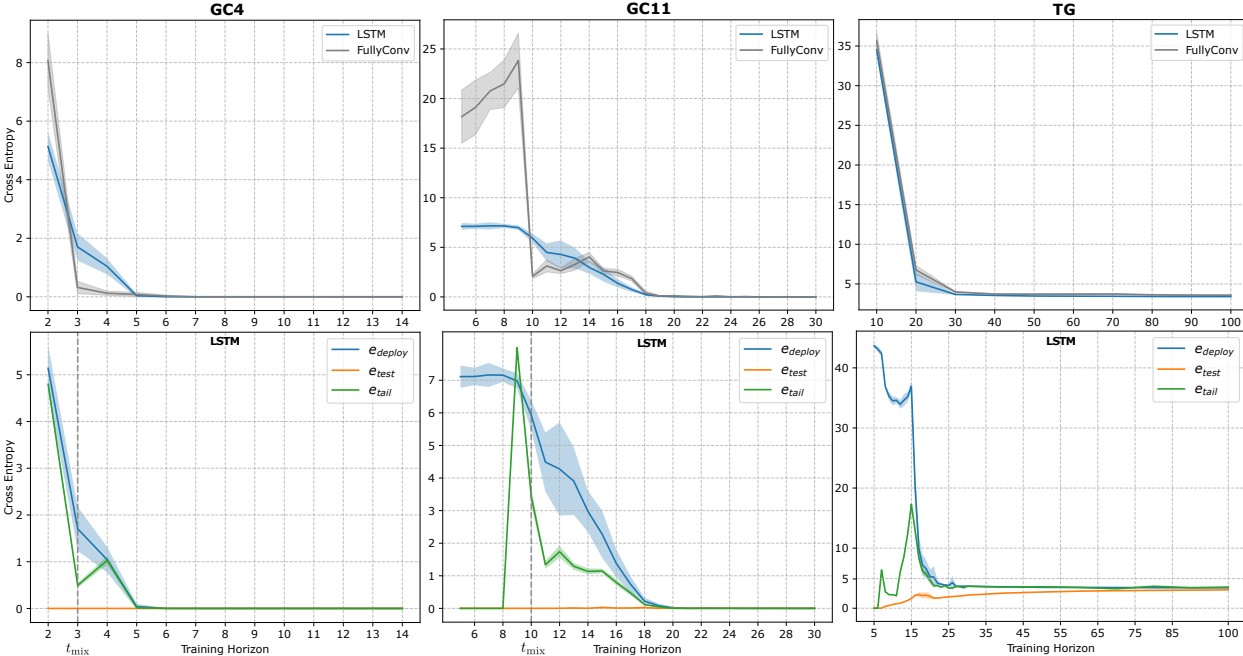

Figure 6: Top panels: deployment error over training horizons for the LTSM and FullyConv network. Bottom panels: deployment, test and tail-test error for increasing training horizon for the LTSM model. The vertical dashed line marks the mixing time of the system, if known. Note that the apparent difference between the blue curves representing the LSTM error in the top and bottom plots is a result of the different scales of the other variables plotted or of different values of the training horizon considered.

**Optimal training horizon.** The top panels of Figure 6 display the deployment errors $e^k_{\text{deploy}}$ of the LSTM and FullyConv models computed for increasing training horizons $h_k \in H$ for the three scenarios. Notably, for GC4 and GC11 the deployment errors drop significantly when $h_{\text{train}} \geq 3$ and $h_{\text{train}} \geq 10$, respectively (see top left and central panels). Clearly, this drop correlates with the mixing times of the systems that correspond to $t_{\text{mix}} = 3$ for GC4 and $t_{\text{mix}} = 10$ for GC11. Essentially, both learning models exhibit improved accuracy as soon as some experience of the steady state is recorded in the training trajectories. However, to achieve error close to zero a few additional training time steps are needed, precisely $h_{\text{train}} \geq 6$ for CG4 and $h_{\text{train}} \geq 20$ for CG11.

To better understand the impact of the mixing time on the model learning, we compare the deployment and test errors progression over training epochs of the LSTM for different training horizons on the CG4 domain in Figure 7. Before the mixing time for $h_{\text{train}} = 2$, no experience of the steady state is stored in the training set. Thus, all the predictions are solely based on the influence experienced in the first 2 time steps. As a consequence, Figure 7(a) shows an overfitting effect: the test error improves over the training epochs and tends quickly to zero as the approximations are very accurate for the first time steps; on the other hand, the deployment error becomes larger for increasing training epochs. After mixing, for $h_{\text{train}} = 4$ the deployment error has a significant decrease. Yet Figure 7(b) shows the same overfitting effect for more training epochs. The reason is that the experience of stantionary influence in the training sequences is still very limited and not sufficient for the learning model to generalize it over the deployment horizon. For $h_{\text{train}} = 6$, Figure 7(c) shows that the deployment error quickly tends to zero together with the test error. Moreover, no significant differences are detected for additional training time steps (see Figure 7(d) for $h_{\text{train}} = 8$). This motivates the choice of optimal training horizon as $h_{\text{train}}^* = 6$. Similarly for GC11, the training curves in Figure 8 Appendix A show that the deployment error starts to decrease for $h_{\text{train}} = 12 > 10 = t_{\text{mix}}$ to tend to zero when the training horizon is around 22 time steps. More training time steps do not improve significantly the predictions. Thus, we select as training horizon $h_{\text{train}}^* = 22$. In the traffic grid domain, as illustrated in the top right panel of Figure 6, the deployment error of both learning models decreases substantially between 10 and 20 time steps. The model performance improves for training horizon ranging from 20 and 30 time steps. However, increasing further $h_{\text{train}}$ does not lead to performance improvement. This is confirmed by the training curves collected in Figure 9 in Appendix A. Although the mixing time of the traffic grid is unknown, these results suggest that the system reaches a stationary influence between 10 and 30 time steps and $t_{\text{mix}}$ should be chosen within those values.

**Deployment error estimate.** The bottom panels of Figure 6 depict the deployment, test and test-tail errors of the LSTM model for different training horizons. The results from all three scenarios confirm uniformly the theoretical intuition presented in Section 3.3. When $h_{\text{train}} \geq t_{\text{mix}}$, the test-tail error (green) reflects the deployment error (blue) trend much better than the test error (orange). Specifically in the two grab a chair scenarios, we observe that after the mixing time, marked by the dashed vertical line, the test-tail error presents the same descending trend of the deployment error. Contrarily, the test error is always very close to zero, indicating that the model learns very well the influence over the training horizon. For the traffic domain, Figure 6 suggests that after approximately 15 time steps the system reaches the equilibrium. Thus, the deployment error and the test-tail error become very close and decrease quickly to converge to the test error. This analysis shows that the test-tail error provides a good estimate of the deployment error and can therefore be used to assess the performance of the learning model using the training set and to choose the optimal training horizon $h_{\text{train}}^*$ as the smallest horizon that yields a low test-tail error.

### 4.3 Observations, limitations and future work

Our study leads to a number of key observations, but there are also limits on how far the conclusions can reach. Here, we provide a balanced overview of both.

**Key observations.** The results of our empirical study show that, even in many complex real-life situations, the task of learning influence can often be quite simple and computationally manageable. This was especially evident in the case of microgrids, where a logistic regression model performs just as well as more advanced models. The complexity of the influence learning problem is affected by the dimensionality and horizon of the forecasting task. For high dimensional or long horizon problems, linear models or vanilla temporal convolutional networks typically fail to provide accurate influence approximations. However, relatively small recurrent and fully convolutional networks have proven to learn good approximations with efficient training times in all domains and thus be the most suitable models for the influence learning task. Additionally, the experiments show that suitable learning models can generalize the influence from a short horizon training trajectories well beyond the training horizon. In general, a good choice for such training horizon depends mostly on the mixing time of the system and it is independent of the learning model. In essence, an optimal training horizon corresponds to the mixing time plus a few additional time steps to collect enough experience

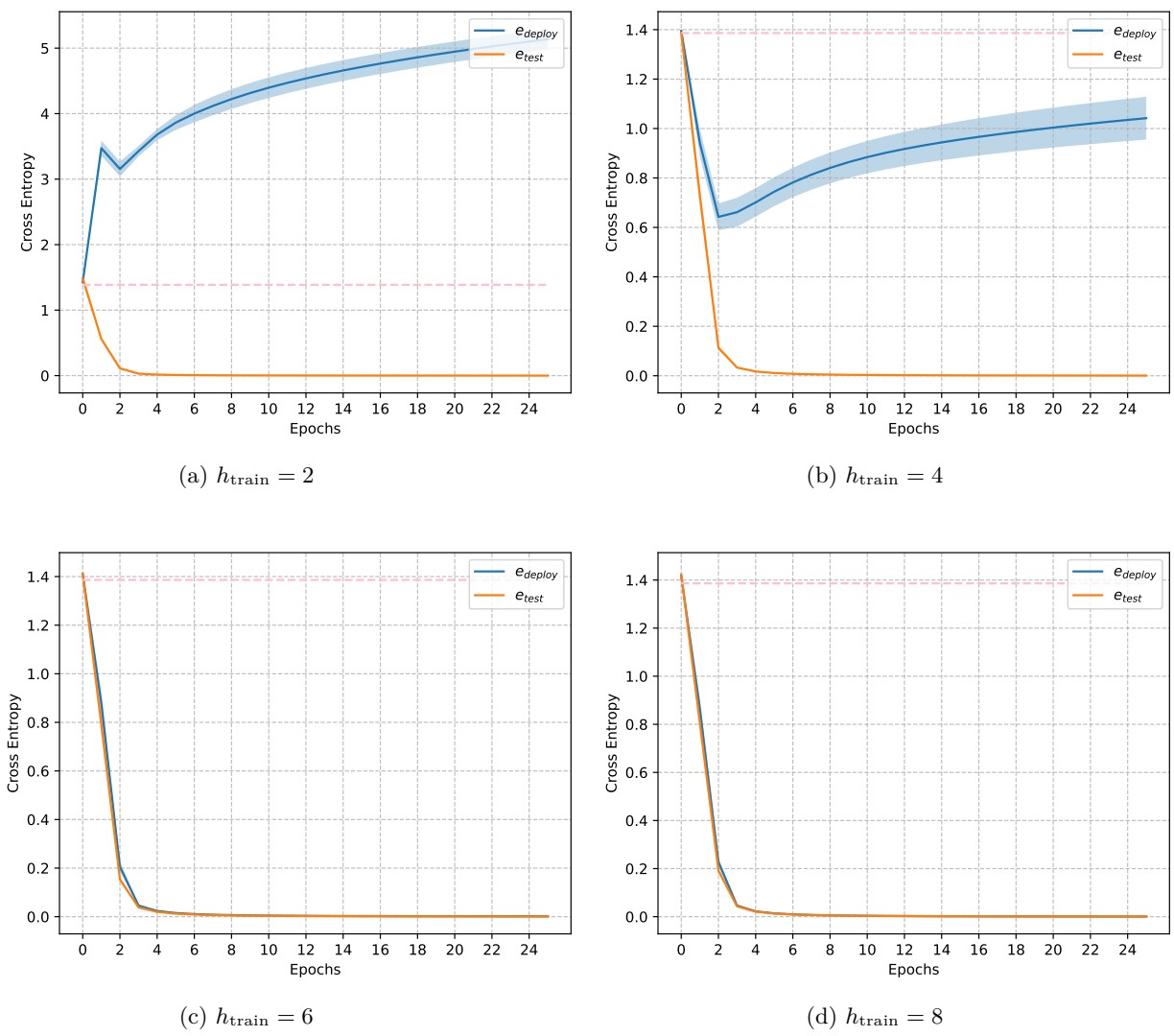

(a) $h_{\text{train}} = 2$

(b) $h_{\text{train}} = 4$

(c) $h_{\text{train}} = 6$

(d) $h_{\text{train}} = 8$

Figure 7: Cross entropy deployment and test errors over increasing training epochs of the LSTM model in the GC4 scenario, computed for different training horizon $h_{\text{train}}$. The dashed line represents the loss of the baseline random classifier.

of the stationary influence in the training set. The test-tail provides a better estimate of the deployment error and therefore can be used to assess the quality of the learning models and select an optimal horizon.

**Environment Scalability.** First, while this study has used relatively complex benchmark problems (microgrid with 100s of agents, and traffic control with more than 100 state variables), we did not exhaustively investigate the scalability of the environments. Scalability of influence prediction is mostly dependent on the number of influence sources that need to be predicted. As long as this is limited, the learning and use of influence can scale very well, as for instance demonstrated by Suau et al. (2022a) that employ influence-based abstraction, in parallel, to a traffic control task with 100 intersections. Furthermore, even though the IBA framework can be naturally extended to continuous state variables, our study is limited to discrete state variables, and does not address the more realistic challenge of learning continuous influence sources. Lastly, although the chosen domains are inspired by realistic scenarios, they still represent simplifications of actual real-world cases.

**Specification of local model and d-sets.** Additionally, our approach relies on the manual specification of local states and d-sets, focusing on the feasibility of influence prediction under these conditions. As discussed in Section 2.5, there are trade-offs between the predictability of influence sources and the size of the local model. Future research could explore automated methods for identifying good decomposition choices to optimize the overall performance.

**Transformers and other learning models.** We note that we have not exhaustively compared all possible sequence models. In particular, transformers, (discrete) diffusion models, and structured state space models could be of interest also to scale to higher dimensional influence prediction tasks. While such recent methods have pushed the limits in the large data regime, IBA is motivated by the idea that performing many simulations with a global simulator could be too expensive. Therefore our focus has been on exploring the potential of learning from limited amounts of data ($N = 500$ trajectories). Our study shows that exploring whether, for instance, (small) transformers could be competitive in this regime, or whether they could provide further benefits when problems become even more complex, is promising. However, these explorations are left for future research.

**Impact for control.** In this paper, we explored the feasibility of learning accurate influence predictors, as measured by CE loss. However, the full IBA framework intends to support decision making: in the end we care about the impact of using the resulting models in, e.g., planning, predictive control or RL. Intuitively the more accurate the influence predictors and hence I-ALM, the better results we would expect. Indeed, in prior work Congeduti et al. (2021) did investigate this question, showing theoretically that the CE error leads to an upper bound on value loss and empirically that there is a positive correlation between CE and value loss. Nevertheless, the relation between CE error and value loss is complex, and this alignment may not always hold (Farahmand et al., 2017; Lambert et al., 2020). Specifically, using additional information about rewards or values can lead to tighter upper bounds and thus faster learning (Farahmand et al., 2017), which can have a significant impact in finite sample regimes (Lambert et al., 2020). Therefore, investigating these potential mismatches and adapting influence models to dynamic, interactive environments remains an important area for future exploration.

## 5 Conclusions

In this paper we investigated learning models and techniques for the influence learning task in realistic scenarios. We run an extensive empirical investigation of the performance of different learning models in a variety of domains. We conclude that complex scenarios may still induce manageable influence learning task. We showed that relatively small recurrent models can achieve the same performance levels as state-of-the-art fully convolutional neural networks. Moreover, we explored how to leverage learning models to build local simulators for long horizons using short training trajectories. In particular, we showed that there exists a training horizon which is sufficient to learn good influence approximations for long (or infinite) horizon problems and how to use suitable error metrics to search for such horizon.

Approximate IBA offers a promising direction for managing real-world decision-making problems. However, inaccuracies of influence approximators in high-stakes domains, such as traffic management, energy distribution, and critical infrastructure may lead to unintended consequences including safety risks, economic inefficiencies or biases in decision-making. While this also holds when applying planning and RL techniques without IBA, IBA could potentially introduce further approximation errors. We recommend that future researchers, developers, and practitioners exploring IBA in applications prioritize rigorous validation to address these ethical concerns.

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

# A    Additional results

Figure 8 and Figure 9 show the test and deployment errors of an LSTM learning model for the influence over the different training horizons in GC11 and traffic grid domain, respectively. See experiments in Section 3.3.

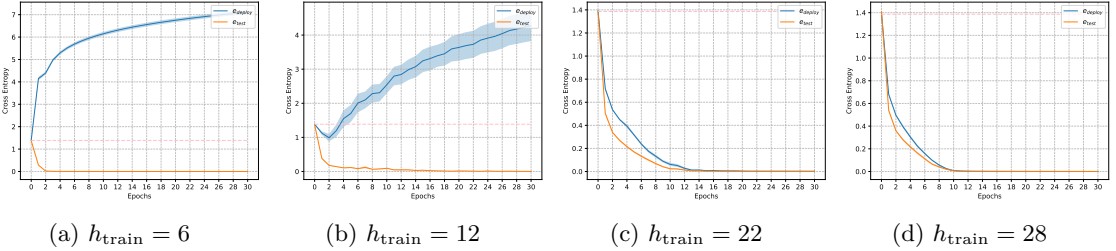

| (a) $h_{\text{train}} = 6$ | (b) $h_{\text{train}} = 12$ | (c) $h_{\text{train}} = 22$ | (d) $h_{\text{train}} = 28$ |

Figure 8: Deployment and test errors for LSTM in GC11 scenario. The dashed line represent the baseline accuracy of the random classifier.

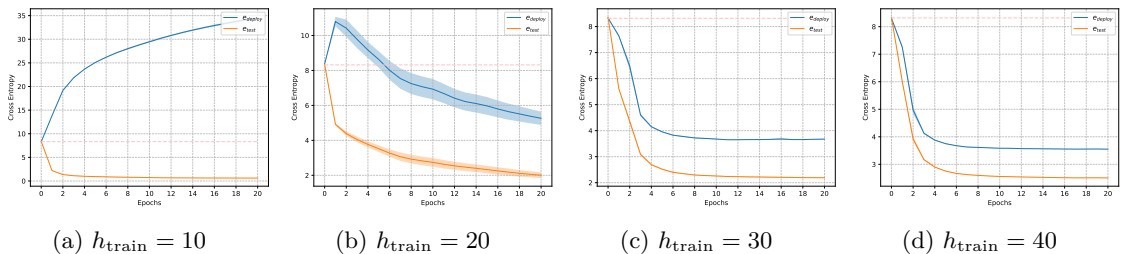

| (a) $h_{\text{train}} = 10$ | (b) $h_{\text{train}} = 20$ | (c) $h_{\text{train}} = 30$ | (d) $h_{\text{train}} = 40$ |

Figure 9: Deployment and test errors for LSTM in traffic grid domain. The dashed line represent the baseline accuracy of the random classifier.

# B  Experimental domains

**Microgrid.**  In this case study, inspired by previous works (Nweye et al., 2023; Li et al., 2012; Vlachogiannis & Hatziargyriou, 2004), we model realistic interactions in an energy district (a microgrid) as a multi-agent problem.  A hundred autonomous agents, referred to as *prosumer units*, manage the energy flexibility of the microgrid. Each agent decides whether to discharge stored energy or trade based on local demand and supply with the goal to minimize costs while ensuring energy balance and independence of the microgrid from the external power grid. Solar and wind energy sources are utilized. The problem focuses on a single unit within a lattice network, with neighboring actions influencing decisions. The task is to predict influence sources based on historical actions. The local states vector is given by, $s_{\text{local}} = (P_{RES}, P_d, SOC)$, where $P_{RES}$ is the renewable power production, $P_d$ the stochastic power demand assumed normally distributed as in Hong & Fan (2016), and $SOC$ is the state of the charge of the battery. The $P_{RES}$ includes solar and wind energy. The former is modeled using hourly solar radiation data in `https://openweathermap.org/api/solar-radiation` and the photovoltaic power generation model introduced by Skoplaki & Palyvos (2009). The wind power generation is calculated by transforming the kinetic energy of the wind speed modelled via the Markov chain model in (Shamshad et al., 2005) and assuming linear relationships with the power produced as in Kuznetsova et al. (2013). The energy produced by an agent may be used to meet the demand $P_d$, or stored in a battery to increase the $SOC$. At any time step $\Delta_t$ (one hour), the agents may decide to discharge the stored energy to meet the demand $P_d$, or trade energy with neighboring units. When buy and sell orders match, the power from the batteries is exchanged at a small cost/revenue for the buying and selling agents respectively. After the local trading is cleared, to satisfy the power balance of single units, every agent is forced to buy residual power an external power distribution gird. A cost $C_{\text{ext}}$ is assigned to

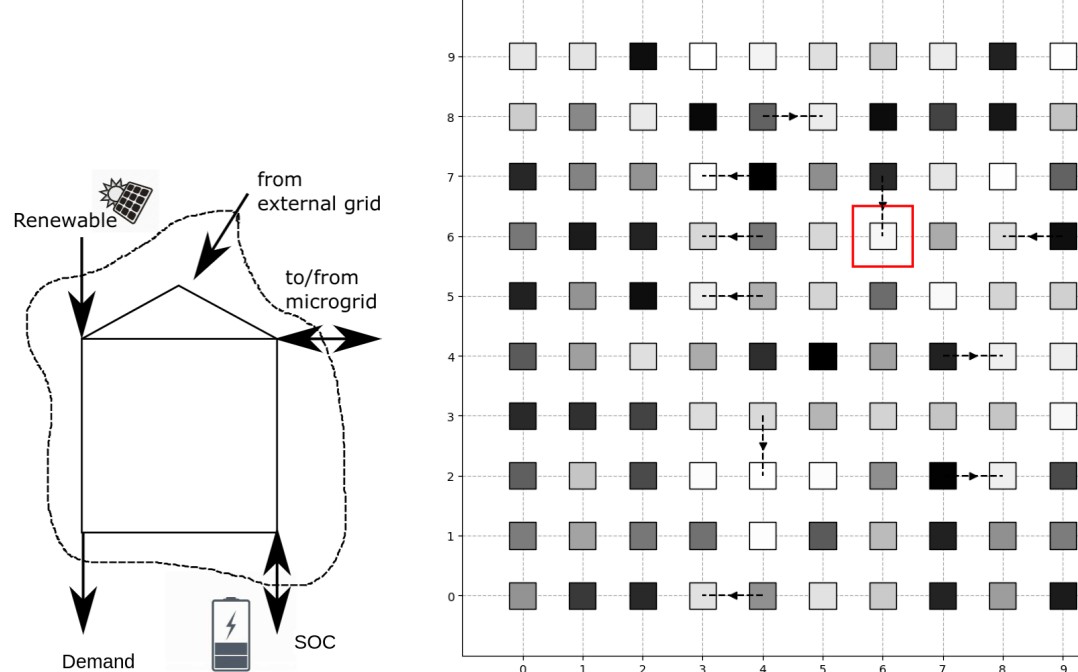

(a) Unit controlled by a single agent in the microgrid.  The local variables observed by the agent include the state of the charge, the power produced by renewable sources and the power demand.

(b) Multi-agent low-voltage grid.  For each unit in the lattice, the state of charge (in percentage) is represented by the gray scale. Directed edges represent a power exchanged between prosumers (agents).

Figure 10: Microgrid.

the energy not supplied ENS $= (P_d - P_{\text{deployed}}) \cdot \Delta_t$, where $P_{\text{deployed}}$ corresponds to the sum of the renewable power deployed and the power discharged from the battery. The cost $C_{\text{ext}}$ for buying from the external grid is much higher than the fixed operational costs of internal trade $C_{\text{int}}$. A schematic representation of a MG unit is depicted in Figure 10(a). The individual reward for each agent is modeled as the sum of the cost/income for the internal trade (if any) and the negative costs of buying the energy not supplied from the external grid (if any) $r = \pm C_{\text{int}} \mathbb{1}_{\text{trade}} - C_{\text{ext}} \text{ENS}$. Thus, the team of agents share the common objective to manage local resources to minimize the electricity costs constrained to satisfying the energy balance, generation limits and storage capacity. We take the perspective of a single unit in a lattice network highlighted in red in Figure 10(b). The initial distribution of the battery is uniformly sampled at random and we consider $h = 40$ hours as the horizon of the problem. We assume that all the other agents act by storing or trying to buy power when the storage is scarce and discharging or trying to sell when power is abundant. Note that besides the problem size can be arbitrarily large, the influence experienced by the local agent only directly depends on the neighboring nodes in the network. Precisely, the only relevant information on the external portion of the system that an agent needs is whether the neighboring north, west, south and east agents will decide to sell or buy power. Besides the distributions of influence sources $s_{src} = (a_{\text{N}}, a_{\text{W}}, a_{\text{S}}, a_{\text{E}})$ are affected (indirectly) by all the agents in the microgrid, the history of local actions $a$ provides a sufficient statistics to predict the influence sources. The resulting problem consists of finding a function approximator for $I(a_{\text{N}}^t, a_{\text{W}}^t, a_{\text{S}}^t, a_{\text{E}}^t \,|\, a^0, \dots, a^{t-1})$.

**Traffic grid.** In this implementation of a traffic network as described in Section 1 and Section 2, we simulate the vehicle traffic in a 9 intersections grid, schematically represented in Figure 11. The sensors of each traffic light capture the vehicles in the $5 \times 5$ local grid at each intersection. The local model is represented as a red square for the selected protagonist agent. The other traffic lights employ hand-coded policies that prioritize lanes with higher car volumes. At time $t = 0$ the grid is initially empty, i.e. the initial state encodes no cars in the network. At any time step, a vehicle will enter the network with a certain probability. The horizon is set to $h = 100$. The state of the environment is represented by binary state variables detecting the presence/absence of a car in a point of the traffic grid. The goal of the agent is to minimize the total number of vehicles waiting at the local intersection. That is, the reward corresponds to the negative number of cars in the local model. To act optimally, the local agent needs to predict if there will be incoming cars from the north end $s_{\text{n}\downarrow}$ and the east end $s_{\text{e}\leftarrow}$. Moreover, the local dynamics is affected by traffic congestion at intersection 2 and 4. In fact, traffic jams can prevent vehicles to move out of the local model from the west and south ends. For this reason, the state variables for the outgoing ends and the actions of agents 2 and 4 are included in the set of influence sources. Thus, in addition to the factors

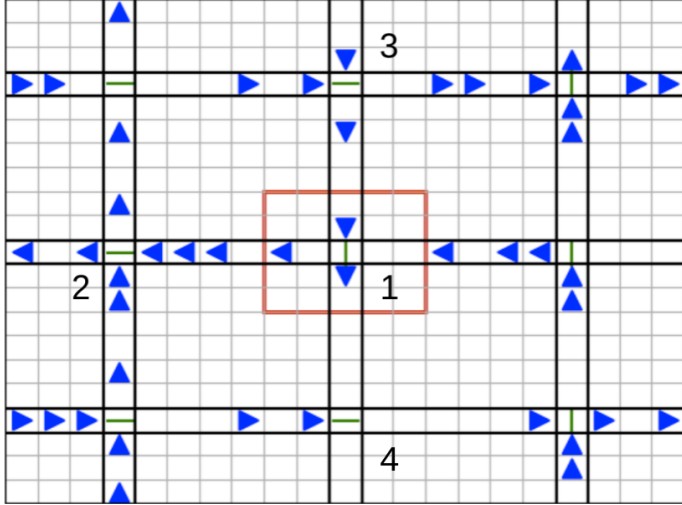

Figure 11: Traffic grid. The local model is delimited by the red square. The blue triangles represent the vehicles and the green bars the traffic lights.

encoding cars inflows, the influence sources encompass 4 variables for the west outflow $s_{w\leftarrow}$, 4 variables for the south outflow $s_{s\downarrow}$, the action $a_2$ and $a_4$, that is $s_{src} = (s_{n\downarrow}, s_{e\leftarrow}, s_{w\leftarrow}, s_{s\downarrow}, a_2, a_4)$. The local information necessary to predict the influence sources includes the entire collection of local variables and actions. The influence that the agent needs to predict is therefore $I(s_{n\downarrow}^t, s_{e\leftarrow}^t, s_{w\leftarrow}^t, s_{s\downarrow}^t, a_2^t, a_4^t \mid s_{local}^0, a_1^0, \ldots, a_1^{t-1}, s_{local}^t)$.

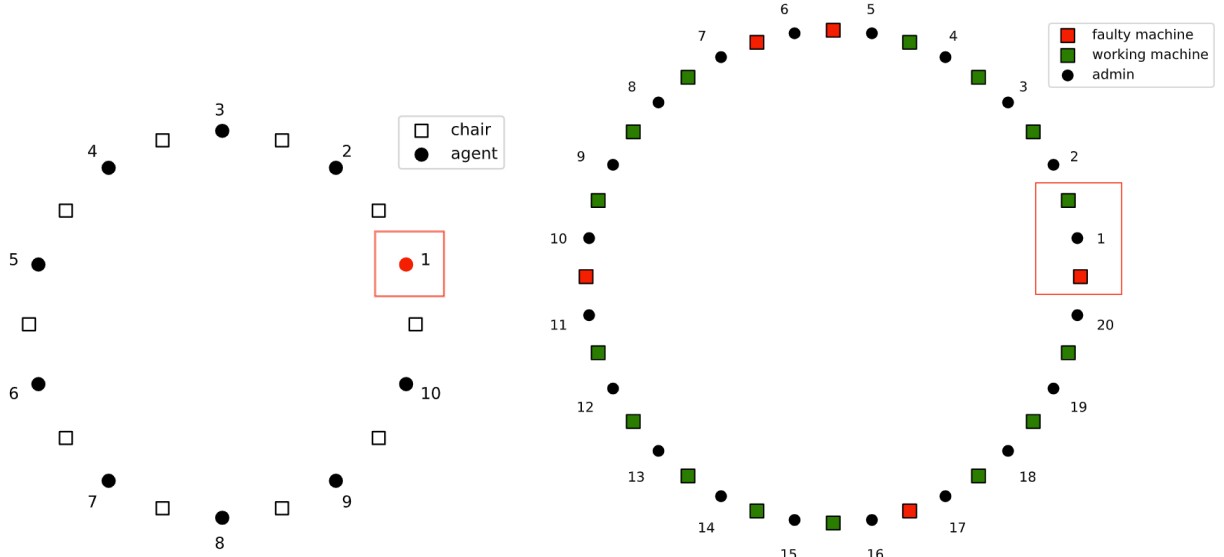

Figure 12: Grab a chair.          Figure 13: System admin.

**System admin.** We use a multiagent version of the System administrator domain from Poupart & Boutilier (2004). A team of system administrators are responsible for the upkeep of a network of machines. Each node has a probability of failing at any time step which increases when a neighboring machine in the network is down. Each agent only observes the status of the machines in its proximity. Consequently it may decide to intervene by trying to reboot the system of one of these nodes. With a certain probability, the process will succeed, resulting in a working node at the next time step. When more than one agent decides to reboot one machine, the process has full rate of success. The goal of the admins team is to secure the highest number of working machines. Precisely, any agent receives a penalty for each faulty machine which lies under its control. We consider a network of $N = 20$ machines organized in a ring configuration as depicted in Figure 13. Each admin agent $i$ is responsible for the maintenance of two neighboring nodes whose states, denoted by $x_i$, $x_{i+1}$, can be fully observed. We take the perspective of a single administrator, for instance agent 1 in Figure 13, whose local model includes only the states (faulty or working) of the two neighboring machines $s_{local} = (x_1, x_2)$ and its action $a_1$. The problem horizon is set to $h = 500$ time steps, and initially a random state is sampled for each machine. To act optimally, agent 1 needs to know if agents 2 and 20 will decide to reboot one of the two machines over which they share the control. Also, it needs to reason about the neighboring machine status $x_3$, $x_{20}$ as they may contribute to the higher chances to turn down the machines in its local model. Then, according to the influence formalism introduced in Section 2, the sources of influence correspond to $s_{src} = (x_3, x_{20}, a_2, a_{20})$. The local information at the disposal of the agent 1 to predict the sources of influence consists of the entire collection of local variables, that is, $(x_1, x_2, a_1)$. Thus, the influence problem consists of finding an approximation for the distribution $I(x_3^t, x_{20}^t, a_2^t, a_{20}^t \mid x_1^0, x_2^0, a_1^0, \ldots, a_1^{t-1}, x_1^t, x_2^t)$.

**Grab a chair.** In this simplified version of the SA problem, introduced by He et al. (2020), $N$ agents disposed in a ring fashion decide at every time step to grab the chair on their left or right side, as shown in Figure 12. They obtain the chair and thus get the reward only if the neighboring agent has not targeted that chair too. After taking an action, each agent only observes whether it managed to grab the chair, ignoring the action of the neighboring agents. The local agent, numbered by 1 and depicted in red in Figure 12, has no access to other information rather than its own actions and rewards which form the local model. The

horizon is set to $h = 200$ and initially every agent chooses deterministically the chair on its right side. After that, all the non-local agents act by copying the previous action of the following agent in counterclockwise order. For the decision making problem of the local agent, the only information required to act optimally consists of the decisions of its neighboring agent 2 and $N$ as they directly influence the possibility to secure a chair. Contrarily, the other agents only affect the local model indirectly. Therefore, the local agent needs to predict the influence sources corresponding to the actions $s_{\text{src}}^t = (a_2^t, a_N^t)$ given the local information of the d-set $d_{\text{set}}^t = (a_1^0, \ldots, a_1^{t-1})$. Therefore the influence to predict corresponds to $I(a_2^t, a_N^t | a_1^0, \ldots, a_1^{t-1})$.

## C    Visualization of the predictions

Short videos showcasing the influence predictions for the traffic grid and grab a chair are available here. In the traffic grid video, the left side displays the global model simulator, while the right side shows the local model alongside the influence source predictions of an LSTM trained over 30 time steps, achieving a cross entropy test error of approximately 3.7 (corresponding to TG in Table 2). The predicted probability distribution over the binary variables, which represent the presence of a car, is visualized using the color intensity: darker colors indicate predictions closer to 1. In this scenario most of the influence sources, precisely those corresponding to the traffic outflows from the local model, present a predominantly deterministic behavior: their values are almost completely determined by the local variables. This allows us to get a visual intuition of the quality of the predictions that seem indeed pretty accurate. In the grab a chair video, we visualize the predictions of an LSTM trained over 6 steps, with the blue markers together with the global simulator. Aligned with the cross entropy test error that is very close to zero (see GC4 in Table 2), the predictions appear to match perfectly the influence source deterministic values.

## D    Implementation details

| Domain | $h$ | #Agents | Policies | $b_0$ | #Influence Sources | D-set dimension |
|--------|-----|---------|----------|-------|--------------------|-----------------|
| Microgrid | 40 | 100 | ranges | uniform | 4 | 1 |
| Traffic grid | 100 | 9 | priority | zeros | 12 | 9 |
| System admin | 500 | 20 | mixed | uniform | 4 | 3 |

Table 3: Setting of the scenarios and features of the influence learning tasks.

| Domain | Optimization | | | | | | |
|--------|-------------|------------|---------|-----|---------|----------|---------|
| | Sample size | Batch size | #Epochs | Alg | Loss | LR Decay | Valid |
| Microgrid | 500 | 100 | 15 | Adam | Entropy | Linear | Split90% |
| Traffic grid | 500 | 100 | 20 | Adam | Entropy | Linear | Split90% |
| System admin | 500 | 100 | 20 | Adam | Entropy | Linear | Split90% |

Table 4: Optimization hyperparameters for the learning models.

| Models | **Architecture** | | | | | |
|---|---|---|---|---|---|---|
| | **#Layers** | **#Units** | **Kernel** | **#Params** | **Activate** | **Regularize** |
| (*size ≤100*) | | | | | | |
| LSTM | 1 | 2 | - | 88 | Tanh | None |
| GRU | 1 | 2 | - | 78 | Tanh | None |
| TCN | 2 | 2,2 | 8 | 100 | ReLU | None |
| FullyConv | 2 | 1 | 8 | 52 | ReLU | Dropout |
| (*size 1000*) | | | | | | |
| LSTM | 1 | 13 | - | 1056 | Tanh | None |
| GRU | 1 | 14 | - | 954 | Tanh | None |
| TCN | 4 | 6,6,6,6 | 8 | 1048 | ReLU | None |
| FullyConv | 4 | 6,6 | 8 | 1084 | ReLU | Dropout |
| (*size 15000*) | | | | | | |
| LSTM | 1 | 56 | - | 14128 | Tanh | None |
| GRU | 1 | 64 | - | 13904 | Tanh | None |
| TCN | 5 | 20,20,20,20,20 | 8 | 13396 | ReLU | None |
| FullyConv | 8 | 15,15,15,15 | 8 | 13246 | ReLU | Dropout |
| LogReg | - | - | - | 13104 | None | None |

Table 5: Microgrid. Architectures of the learning models.

| Models | **Architecture** | | | | | |
|---|---|---|---|---|---|---|
| | #Layers | #Units | Kernel | #Params | Activate | Regularize |
| (*size ≤200*) | | | | | | |
| LSTM | 1 | 2 | - | 176 | Tanh | None |
| GRU | 1 | 2 | - | 150 | Tanh | None |
| TCN | 2 | 2,2 | 4 | 164 | ReLU | None |
| FullyConv | 2 | 2 | 4 | 188 | ReLU | Dropout |
| (*size 1000*) | | | | | | |
| LSTM | 1 | 9 | - | 960 | Tanh | None |
| GRU | 1 | 11 | - | 1014 | Tanh | None |
| TCN | 4 | 4,4,4,4 | 10 | 976 | ReLU | None |
| FullyConv | 4 | 4,4 | 10 | 1032 | ReLU | Dropout |
| (*size 10000*) | | | | | | |
| LSTM | 1 | 42 | - | 9936 | Tanh | None |
| GRU | 1 | 49 | - | 10020 | Tanh | None - |
| TCN | 4 | 16,16,16,16 | 10 | 9592 | ReLU | None |
| FullyConv | 4 | 16,16 | 10 | 9816 | ReLU | Dropout |
| (*size 50000*) | | | | | | |
| LSTM | 1 | 104 | - | 50360 | Tanh | None |
| GRU | 1 | 120 | - | 50064 | Tanh | None - |
| TCN | 6 | [30]x6 | 10 | 48624 | ReLU | None |
| FullyConv | 6 | [30,30,30] | 10 | 49104 | ReLU | Dropout |
| (*size 1M*) | | | | | | |
| LogReg | - | - | - | 1093200 | None | None |

Table 6: Traffic grid. Architectures of the learning models.

| Models | #Layers | #Units | Kernel | #Params | Activate | Regularize |
|--------|---------|--------|--------|---------|----------|------------|
| | | | **Architecture** | | | |
| *(size ≤100)* | | | | | | |
| LSTM | 1 | 2 | - | 80 | Tanh | None |
| GRU | 1 | 2 | - | 66 | Tanh | None |
| TCN | 2 | 2,2 | 4 | 68 | ReLU | None |
| FullyConv | 2 | 2 | 4 | 80 | ReLU | Dropout |
| *(size 1000)* | | | | | | |
| LSTM | 1 | 12 | - | 920 | Tanh | None |
| GRU | 1 | 14 | - | 918 | Tanh | None |
| TCN | 4 | 6,6,6,6 | 8 | 1088 | ReLU | None |
| FullyConv | 4 | 6,6 | 8 | 1136 | ReLU | Dropout |
| *(size 10000)* | | | | | | |
| LSTM | 1 | 48 | - | 10568 | Tanh | None |
| GRU | 1 | 54 | - | 9998 | Tanh | None |
| TCN | 6 | [16] x6 | 8 | 10856 | ReLU | None |
| FullyConv | 6 | 16,16,16 | 8 | 11016 | ReLU | Dropout |
| *(size 3M)* | | | | | | |
| LogReg | - | - | - | 2.9M | None | None |

Table 7: System admin. Architectures of the learning models.

| Domain | $H$ | $h_{\text{deploy}}$ | #Agents | Policies | $b_0$ |
|---|---|---|---|---|---|
| GC4 | $\{2, \ldots, 14\}$ | 200 | 4 | copy | Deterministic - right chair |
| GC11 | $\{5, \ldots, 30\}$ | 200 | 11 | copy | Deterministic - right chair |
| TG | $\{10, \ldots, 100\}$ | 500 | 9 | priority | Deterministic - zero vehicles |

Table 8: Setting of the scenarios for long horizon tasks.

| Domain | Batch size | #Epochs | LR Init | LR Final | LR Decay | Train size $n$ | Test size $m$ |
|---|---|---|---|---|---|---|---|
| GC4 | 10 | 25 | $10^{-2}$ | $10^{-5}$ | linear | 500 | 100 |
| GC11 | 10 | 30 | $10^{-2}$ | $10^{-5}$ | linear | 500 | 100 |
| TG | 10 | 20 | $10^{-2}$ | $10^{-5}$ | linear | 500 | 100 |

Table 9: Optimization choices for long horizon tasks.

| Domain | Model | Architecture | | | | | |
|---|---|---|---|---|---|---|---|
| | | #Layers | #Units | Kernel | #Params | Activate | Regularize |
| GC4 | LSTM | 1 | 10 | - | 564 | Tanh | None |
| | FullyConv | 4 | [6,6] | 4 | 544 | ReLU | Dropout |
| GC11 | LSTM | 1 | 32 | - | 4612 | Tanh | None |
| | FullyConv | 8 | [10,10,10,10] | 6 | 4484 | ReLU | Dropout |
| TG | LSTM | 1 | 32 | - | 2136 | Tanh | None |
| | FullyConv | 4 | [8,8] | 6 | 1944 | ReLU | Dropout |

Table 10: Architectures of the learning models for long horizon tasks.

