# OpenReview forum: "Influence Learning in Complex Systems"
_TMLR — Accepted by TMLR_

### Review · Reviewer_fDcn · 2024-11-11

**Summary Of Contributions:**

The paper presents an overview of the problem of learning local influence models to approximate large complex factorized systems. The authors present an introduction to the problem and a thorough overview of different learning approaches for predicting the evolution of local systems based on approximating the unobserved influences of other actors.

**Audience:**

Yes

**Claims And Evidence:**

Yes

**Requested Changes:**

Please evaluate some non-recurrent baselines.

Please comment on the questions.

**Strengths And Weaknesses:**

### Strengths
The paper is overall very straightforward and easy to follow. The problem, its relevance, and the general approach is well motivated and clear. The empirical comparisons are well set up and thorough. As far as I can tell, experiments seem to have satisfactory statistics robustness.

### Weaknesses
The main missing component of the work are twofold for me:
- Baseline comparisons. A significant component of the motivation for the tested algorithms is that they can aggregate temporal information to improve POMDP prediction. To clearly show this, I think the authors should present a simple non-recurrent model as a baseline to show _how_ important history aggregation is.
- Continual learning and control: A substantial amount of motivation for the problems is control. However, all experiments are evaluated in prediction. It is relatively well established that superior predictive performance does not always automatically translate to better controllers (compare e.g. [1]). The paper seems somewhat incomplete to me without evaluating the models in predictive control or RL. In addition, moving to the control case will require adapting to changing influences as the agent's decision influences that of agents around them. As many standard ML approaches fail to perform robustly in a continual learning setup such as the one encountered in many RL problems, I think this would be a vital test. In this scenario, the assumption of mixing might also break, as agents interacting might create more complex long range dynamics.

### Question
The overall takeaway from the paper is slightly unclear to me. Are the used environments important tasks for the community that focuses on these problems? If not, I think it would be important to add benchmarks of the complexity that a practitioner would recognize as hard or important. As I am not a domain expert here, it might well be that this is already the case. In any case, I'd be grateful if the authors commented on this and clarified this in the paper. An empirical claim about learnability is always somewhat contingent on whether a problem is interesting as a representation of a practically relevant problem (in my opinion), or at least a problem that a substantially large enough research community has agreed to care about.

While previous reviews apparently called for better formalization, I don't see how the specific setup of an influence setup is used here. I would assume that one of the advantage of using a factorized model is that learning agents can be reused over all components. But by only taking a local view here, I don't really see how the problem differs from a regular POMDP? My question is: what parts of the evaluated approaches are specific to influence learning, and which ones just hold for POMDPs? Does the formalization allow us to make some assumptions on the influence that simplify the learning problem?

### Smaller remarks
Transformers are not significantly more complex or necessary larger than LSTMs. They are used as components in massively large systems, but training a small transformer is definitely not significantly more difficult than training a small LSTM.

[1] Objective Mismatch in Model-based Reinforcement Learning, Lambert, N. et al., http://proceedings.mlr.press/v120/lambert20a/lambert20a.pdf

---

> ### Author Response · Authors · 2024-11-30
>
> We thank the reviewer for the comments and questions.
>
> **Weaknesses**
> - (Baseline comparison) We agree on the importance of comparing against a simple non-recurrent model and for this, our paper already includes a logistic regression model as a baseline for comparison with the recurrent models. The performance of this baseline is reported in Table 1, Figure 4 and 5, and discussed in Section 4.1 to conclude that, in high dimensional or long horizon problems, linear models typically fail to provide accurate influence approximations (Section 4.3).
>
> - (Continual Learning and Control) We agree that evaluating the proposed models in a control framework would be an interesting extension of this work. However, addressing control challenges comprehensively—including adapting to dynamically changing influences, potential violations of the mixing assumption, and ensuring robustness under continual learning conditions—would broaden the scope of the current paper significantly, necessitating a separate, in-depth study.
> To address the reviewer’s concern about whether improved predictive accuracy translates to better control performance, we highlight that the work Congeduti et al, AAMAS (2021) presents theoretical and empirical evidence that minimizing average cross-entropy of the influence predictions aligns well with improving the control solutions in some IBA settings. However, we acknowledge that prediction quality does not universally guarantee superior control performance, as highlighted in Lamber et al. [1] and Farahmand et al. [2] before that. In the revised manuscript, we will reference both papers and include the following clarification:
>
>  "As a direction for future research, evaluating the performance of influence models in control frameworks, such as predictive control or reinforcement learning, could provide valuable insights. While improved predictive accuracy often aids control performance, as supported by prior results [AAMAS 2021], this alignment may not always hold universally [1,2]. Specifically, while model accuracy leads to an upper bound on value loss [2, AAMAS 2021],  using additional information about rewards or values can lead to tighter upper bound and thus faster learning [2], which can have a significant impact in finite sample regimes [1].  Investigating these potential mismatches and adapting influence models to dynamic, interactive environments remains an important area for future exploration."
>
> [2] Farahmand, Amir-massoud, Andre Barreto, and Daniel Nikovski. "Value-aware loss function for model-based reinforcement learning." Artificial Intelligence and Statistics. PMLR, 2017.
>
> **Questions**
> - This study explores "influence-based abstraction (IBA)" applied to practical domains. IBA is a specific form of abstraction and therefore, somewhat a niche topic.  However, given the potential of this approach to significantly speed-up planning and reinforcement learning, it has made an impact at major venues such as NeurIPS, ICML, and IJCAI. Our goal in this paper is to assess the capacity of current machine learning techniques to learn accurate, approximate influence representations in benchmarks that are closer to real-world conditions (e.g., large-scale problems with many influence sources and long deployment horizons). Thus, while we do not claim yet a direct impact on specific real-world tasks, this work demonstrated the potential scalability of this approach to address complex, real-world challenges.
> We will include one paragraph to highlight the practical relevance of the considered environments.
> - IBA is a framework to exploit structure and enable a local agent, in a factored POMDP, to do planning or RL in a smaller 'local' model, that is also a POMDP. Given that such local decision problems frequently happen in large spatial tasks, IBA is often explored in decision-making problems that could have many agents (e.g., traffic control, power grids). However, the concept of learning influences, indeed, revolves around a single agent's capability of predicting its local next state. The problem does not differ from a regular factored POMDP.
> The factored structure can be used to express additional simplifying assumptions (see Section 7 of Oliehoek et al. 2021,JAIR) leading to tractable (more compact) influence representations. In most cases such assumptions are too strong. However, it is not unlikely that the factored structure itself, without additional assumptions like transition independence, may still result in particular influence patterns. In this case, we expect machine learning methods to perform well in learning influence approximations. In this paper, we give evidence that indeed simple ML models can capture these influence patterns.
>
> **Smaller Remark**
>
> We agree with the reviewer’s point. A small transformer could perform competitively, and potentially even outperform the models we examined. We will include this in the revised manuscript as an interesting direction for future research.

---

> ### Author Response · Authors · 2025-01-26
>
> To incorporate all the points raised by the other reviewers, we have included in the manuscript this slightly different version of the paragraph on control in Section 4.3
>
> “In this paper, we explored the feasibility of learning accurate influence predictors, as measured by CE loss. However, the full IBA framework intends to support decision making: in the end we care about the impact of using the resulting models in, e.g., planning, predictive control or RL. Intuitively the more accurate the influence predictors and hence I-ALM, the better results we would expect. Indeed, in prior work Congeduti at al [AAMAS21] did investigate this question, showing theoretically that the CE error leads to an upper bound on value loss and empirically that there is a positive correlation between CE and value loss. Nevertheless, the relation between CE error and value loss is complex, and this alignment may not always hold [1,2]. Specifically, using additional information about rewards or values can lead to tighter upper bounds and thus faster learning [2], which can have a significant impact in finite sample regimes [1]. Therefore, investigating these potential mismatches and adapting influence models to dynamic, interactive environments remains an important area for future exploration.”

---

### Review · Reviewer_sXHj · 2025-01-12

**Summary Of Contributions:**

The Influence Learning Task (ILT), as described in that paper, consists in solving complex multi-agent POMDP by focusing on local dynamics, and estimating the influence (conditional probability) of the past local states and actions over the current local state.

As the time horizon of the global POMDP is not necessarily known or is supposedly large, the learning is performed over a set of trajectories of limited and fixed horizon (h_train).

The paper explores the ability of simple models (like LSTM or GRU) with different sizes, as well as the impact of the training horizon, at estimating the influence. Specifically, it focuses on two difficulties: large dimensions of the global POMDP, and large horizon (h_deploy >> h_train).

**Audience:**

Yes

**Broader Impact Concerns:**

No concern

**Claims And Evidence:**

Yes

**Requested Changes:**

The paper could significantly gain from an interpretation of the obtained CE losses.
This could be given by some reward scores given a policy based on the different obtained influence estimators.
Or maybe (if possible) giving some indicator threshold of the sufficient CE levels to solve the task.

Other minor details and questions:

1) I'm not sure to understand why the history of local variables is enough to determine the source variable: doesn't it also depend on the history of all external variables (for ex, past actions of other agents) ?

2) Just above eq.2:  Î(S_src | ...) instead of Î(S_srt | ...)

3) Ch3, paragraph 1: "humans are generally quite successful in ..." --> can you provide a reference, or at least an example to support that claim?

4) in Eq 5: IIUC, in order to force the induced Markov chain to be ergodic, the policy of all agents is determined by a single agent: a_i^t = a_1^t-i+1, and this policy is used as an explorative one to get the database of transition to learn the influence. But this let apart all possible transitions where the agents are acting independently?

5) Eq 8: IIUC the left-hand term in the error is the one posing a problem, since it is computed before the dynamics closely reach a stationary distribution. Then, how does the tail error that starts summing at h_k and not at t_mix remove that term? Do we have h_k > t_mix?

6) Fig 6, TG task: How is it possible that LSTM + e_deploy in the top curve can differ from the same in the bottom curve ?

**Strengths And Weaknesses:**

Strengths

The paper is well written, easy to follow and provides a good introduction to the problem.
The contribution is clear: this is about testing various models at solving ILT in realistic scenarios.
The experimentation is rich enough (many architectures, sizes and environments, 10 seeds), is well described, and answers the asked questions.

Weaknesses

While the different models are well compared together, one missing point is an interpretation of how good they are at solving the task.
For example, looking at LSTM on the traffic grid (in fig. 3), how do we interpret a cross-entropy loss reaching ~3.5 ? is it good ? Does it significantly help at solving the underlying RL task ?

---

> ### Author Response · Authors · 2025-01-26
>
> We really appreciate the detailed comments and interesting questions raised.
>
> **(CE loss interpretation)** thank you for the interesting observation that CE is useful for relative comparison, but difficult to interpret absolutely.
> In some special cases, we are able to provide this interpretation. In grab a chair, the true influence distribution is deterministic. Then, the target CE for a perfect model is 0. Figure 6 and 7 show that the models learn approximately perfect influence, CE~0. In case of perfect approximation, we can also conclude that there is no loss in value for the control problem, as shown by Congeduti et al. [2021]. For the other realistic scenarios, we do not have such strong assumptions. However, for a more concrete idea of the quality of the predictions, this short simulation https://surfdrive.surf.nl/files/index.php/s/EsAU2fPWzPC8bYh shows the traffic grid simulator together with the predictions of the LSTM corresponding to TG in Table2 . The video shows that the model (CE ~ 3.7) achieves good influence approximations. We have included this video and a video for GC4 in AppendixC.
> More generally, this is a difficult question. The interpretation of CE errors requires some knowledge on the true influence entropy (Bayes error). Estimating or bounding the Bayes error of a probabilistic model is generally not a trivial task [Q. Chen et al., IEEE PAMI (2023)]. Nonetheless, CE error is still often considered the default metric for evaluating probabilistic models, e.g. generative models [L. Theis et al., ICLR (2016)].
>
> **(Relation CE loss - value loss)** We agree that it would be very interesting to investigate the impact of different influence-approximation errors on the loss in the achieved value. In previous work, Congeduti et al. 2021 examined this question both theoretically and empirically. They study the correlation between CE and value loss in toy problems where it is possible to compute the exact optimal policy and value. The empirical results in Figures 2, 4 of that paper indicate that there is a clear correlation between CE error and value loss. We highlight that one of the simulation domains (figure 4) is a smaller version of the traffic grid. However, we would like to stress that comparisons like this are inherently limited to small-scale problems (much smaller than we consider in our work): computing the exact optimal policy both for the exact model, as well as for a specific influence approximation is possible only in very small scale domains. Another option would be to compare approximate policies (e.g., learned with RL), obtained for local models corresponding to different influence estimators, as suggested by the reviewer. However, this would introduce a different source of error on the value which could not be disentangled. We would only be able to observe the joint effect of influence and policy approximations. Given that the relation between influence approximation and optimal value loss is hard to interpret (as the reviewer correctly points out), such a compound error would be even more difficult to interpret. We have added a paragraph on this in 4.3
>
> 1.This is what IBA theory shows, specifically,
>
> LH = local state history / d-set (hidden)<br>
>
> GH = global state history (hidden)
>
> OH = action-observation history (observed)
>
> then:
>
> P(u | OH) = \sum_{LH} P(u | LH) P(LH|OH) {since LH d-separates the influence sources from OH)
>
> where
>
> P(u| LH ) = \sum_{GH} P(u | LH, GH) P(GH|LH)
>
> does not depend on OH {since LH d-separates the influence sources from OH, it also d-separates from GH).
>
> 3.This is more of an intuitive idea rather than a claim, based on the observation that large and complex problems are generally not solved in a centralized fashion but are instead broken down into smaller, more manageable sub-tasks. One example is the “division of labour”, a core principle in economics and organizational theory that highlights how organizations achieve efficiency with the separation of the tasks. Also, in traffic management, the current traffic control practitioners focus on areas of interest, drawing boundaries that enable them to do some amount of useful optimization. We have revised the sentence to underline that we express an intuitive idea.
>
> 4.The external policies are features of the experimental domains: different choices induce different influence distributions. For the specific scenarios mentioned (GC4-11), the specific choice of the external policies ensures to have controlled problems where we know the exact expression of the influence (Eq5) and stationary influence. This allows us to verify exactly whether the learning models provide accurate approximations. To assess more realistic situations where we do not have such knowledge and agents act independently, we use the traffic grid domain. We have emphasized the motivation for this choice in 3.3

---

> > ### Author Response · Authors · 2025-01-26
> >
> > 5.Indeed, the terms before mixing may deviate from the error after mixing. In general, we do not know the exact mixing time of a Markow system, and estimating it can be quite challenging [Wolfer,et al.PMLR (2019)]. However, we can empirically test different increasing values of the training horizon h_k, assuming that for values greater than the mixing time, the “tails” of the test error (eq9) are close enough to the deployment error. We have now clarified this in 3.3
> >
> > 6.They are the same: the apparent differences are a result of the scales of the other variables plotted (GC) and different values of the training horizons (TG). We have included a sentence to the caption to warn the reader for the potential confusion.

---

### Review · Reviewer_A5tQ · 2025-01-21

**Summary Of Contributions:**

The paper investigates the problem of influence-based abstraction (IBA) in complex systems, focusing on accurately approximating the influence of external factors and other agents' policies on local system dynamics. IBA decomposes large, computationally expensive systems into smaller local models by abstracting away non-essential factors while retaining key influences, enabling more efficient reinforcement learning (RL) and planning. The authors conduct an empirical evaluation of existing learning methods, such as LSTM, GRU, and Temporal Convolutional Networks (TCN), across real-world-inspired domains like traffic control, power grids, and system administration tasks and game like grab a chair. The results of the empirical study show that, even in many complex real-life situations, the task of learning influence can often be quite simple and computationally manageable. This was especially evident in microgrids, where a logistic regression model performs just as well as more advanced models. The complexity of the influence learning problem is affected by the dimensionality and horizon of the forecasting task. For high-dimensional or long-horizon problems, linear models or vanilla temporal convolutional networks typically fail to provide accurate influence approximations. However, relatively small recurrent and fully convolutional networks have proven to learn good approximations with efficient training times in all domains and thus be the most suitable models for the influence learning task. Additionally, the experiments show that suitable learning models can generalize the influence from short-horizon training trajectories well beyond the training horizon. In general, a good choice for such training horizon depends mostly on the mixing time of the system and is independent of the learning model. In essence, an optimal training horizon corresponds to the mixing time plus a few additional time steps to collect enough experience of the stationary influence in the training set. The test-tail provides a better estimate of the deployment error and therefore can be used to assess the quality of the learning models and select an optimal horizon. The paper concludes that IBA is a practical approach for decision-making in complex real-life systems, offering a scalable and computationally efficient solution for RL and planning applications.

**Audience:**

Yes

**Broader Impact Concerns:**

The paper does not explicitly include a Broader Impact Statement, and while the topic primarily focuses on influence-based abstraction in reinforcement learning for complex systems, there are potential ethical concerns that should be addressed. One key concern is the applicability of the proposed methods in high-stakes decision-making domains, such as traffic management, energy distribution, and critical infrastructure, where inaccurate influence approximations could lead to unintended consequences, such as safety risks or economic inefficiencies. The authors should discuss potential risks associated with relying on approximate models in these scenarios, emphasizing the importance of rigorous validation before deployment in real-world settings.

Additionally, the paper should address the potential biases introduced by model approximations, particularly when dealing with incomplete or imbalanced data. If the influence learning models fail to capture certain system dynamics accurately, it could lead to unfair or suboptimal decision-making, disproportionately affecting certain groups or regions in applications like urban planning or resource allocation.

**Claims And Evidence:**

Yes

**Requested Changes:**

1. some figures need a better caption to better clarify what the figures mean and try to tell the reader. For example, figure 5's caption is too limited in its explanations and need to clarify what are the different curves in the figure and what do they try to show.
2. explanation why the current selection of hyperparameters make sense. For example, why choose LSTM of 1 layer and not more for the traffic grid problem or system admin problem?
3. try to include more baselines, such as traditional RL based approach, or simpler rule-based approach to show the added value of using machine learning for influence learning.
4. The concept of the d-set, while introduced in the theoretical framework, lacks sufficient discussion on its practical implications. Clarifying how different choices of d-sets impact influence approximation accuracy would enhance the practical utility of the approach.

**Strengths And Weaknesses:**

The paper presents a well-executed empirical study on influence-based abstraction (IBA), offering valuable insights into how influence approximations can be effectively learned and applied to complex systems. A major strength of the paper lies in its comprehensive set of experiments conducted across diverse real-world-inspired domains, such as traffic control, power grids, and system administration, demonstrating that influence learning is often computationally manageable even in large-scale environments. The findings provide practical and actionable conclusions, showing that small-scale models like LSTM and TCN can achieve accurate approximations without the need for complex architectures, making the approach suitable for resource-constrained applications. The analysis of training horizon selection and its dependence on system mixing time presents a useful heuristic for optimizing learning pipelines, balancing computational cost and accuracy effectively.

However, the study has several limitations that could be addressed to enhance its impact. The scope of the experiments is limited to relatively small-scale problems, leaving questions about the scalability of the approach to larger and more complex environments. Additionally, the paper focuses solely on discrete state variables, which restricts its applicability to many real-world systems that involve continuous state spaces. The reliance on older network architectures, such as LSTM and TCN, rather than state-of-the-art models like Transformers, may limit the direct applicability of the findings to modern AI systems, where Transformers are increasingly favored for their superior long-range dependency modeling and scalability. Furthermore, while the paper provides valuable guidance, it would benefit from a broader discussion on how the proposed methods compare with existing work in the field and whether similar conclusions have already been established in prior research.

---

> ### Author Response · Authors · 2025-01-26
>
> We thank the reviewer for the comments. Here we clarify how we have improved the paper according to the points and questions raised.
>
> **(Scalability)** We agree with the reviewer that investigating scenarios with a higher number of (potentially continuous) influence sources would be interesting. We want to highlight that, although we focus on relatively small-scale scenarios with discrete variables, as such MicroGrid (100 agents, >300 state variables) and TrafficGrid (9 agents, >100 state variables), the size of these problems is already large enough to prevent the use of planning methods for optimal control solutions. Moreover, we show how the scale of the problem does not affect the scale of the influence learning task, as the number of influence sources to predict remains quite small independently of the number of agents or state variables. We have now restructured the limitation and future work Section 4.3 to highlight this argument.
>
> **(State-of-the-art models)** Sequence modeling is a popular topic in ML, and next to transformers there are now diffusion models, MAMBA, etc., all of which could be explored for their potential to also deal with the influence prediction task. As such, we agree that transformer models might be suitable or even outperform the models considered for the influence learning task. We have mentioned this in Section 4.3 as an interesting direction for future research.
>
> **(Requested changes)**
> - We appreciate the reviewer’s suggestion for improving the paper readability. We have refined the captions of Figure 4, 5, 6, 7 to enhance clarity.
> - The hyperparameters defining the network architectures have been chosen to ensure a fair comparison between learning models with the same network sizes. Since our goal is to prove that smaller and simpler models could achieve satisfactory performance, we have tested relatively shallow networks. A small number of layers is also preferable as deeper networks would also require a larger amount of global simulator trajectories to be properly trained [Bengio et all, 2013]. However, the choice of the depth also depends on the model class. Specifically, the recurrent models (LSTM, GRU) with only one layer achieve better performance than all the other models (in GrabAChair, near-perfect approximation, CE~0). Thus we decided not to explore deeper architectures. On the other hand, convolutional-based models are known to benefit from deeper architectures to ensure both better performance [He et all (2016), LeCun et all, (1998)]  and full receptive field coverage. We have clarified this point in Section 3.2.
> - Since in this paper (but please see the updates discussion in Section 4.3) we do not address the control problem, we cannot use RL approaches as baselines for the models. Also, we are uncertain about how to manually design rule-based baselines for the influence learning task. Only in cases where we would already know the influence source values in advance (e.g. in grab a chair with deterministic influence) or under other strong assumptions that could be possible. However this comparison would only reveal information about the efficacy of inference methods to deal with such (overly) specific cases. Therefore, our approach is to use logistic regression as a baseline for the recurrent approaches and we report the performance of a random classifier (Figures 4, 7) which is a common general baseline for sequence-to-sequence prediction tasks.
> - The d-set is a set of local variables needed to predict the influence sources: additional variables do not add any information on the influence sources. In other words, the influence sources are conditionally independent of any other local variables v1, v2, v3 ...,
> P(s_{src} | d-set) = P(s_{src} | d-set, v1, v2, v3...).
> In general, there might be multiple sets that satisfy this d-separation property and indeed the complexity of the influence learning task may depend on the choice of the d-set. In practice, we choose a minimal d-set as this is also the best choice for the set of input features of the learning models: using a larger (still d-separating) set of local variables will not improve the accuracy of the learning models. Instead, it would increase the dimensionality of the input of the learning problem, generally making learning more difficult. We have sharpened the definition of the d-set in Section 2.4 and the discussion on different choices of local variables and dset in Section 2.5 to avoid the confusion.
>
> **(Broader Impact Concerns)** In our experience, such in-depth impact statements are rarely included in papers in RL that focus mainly on methodologies rather than direct applications. However, we appreciate the reviewer ‘s detailed suggestion to reflect on the impact of our work and we have included one paragraph in the Conclusions on ethical concerns.

---

### Author Response · Authors · 2025-03-24
**Uploaded camera ready version**

We would like to thank the reviewers and editors for the insightful discussion and the time spent for reviewing our manuscript. We are grateful for the opportunity to publish our work at TMLR and we have uploaded a camera ready version.

---

### Decision · Action_Editor_Wv2M · 2025-03-09

**Recommendation:** Accept as is

**Comment:**

The reviews raised several questions about the scope of the empirical evaluation that were addressed in the rebuttal.  Two out of three reviewers recommended acceptance while the third reviewer remained concerned by the lack of experimentation with continuous and control problems.  Since the current work does make a valuable contribution in the context of inference for large problems with discrete variables, this work still deserves to be published.

**Audience:**

This work is of interest to practitioners working on large scale inference problems in traffic, microgrid and network applications.

**Claims And Evidence:**

The paper studies the application of influence-based abstraction to decompose large scale models into approximate local models.   The paper claims that such approximate local models can be obtained in large scale traffic, microgrid and network inference problems.  An extensive empirical evaluation provides good evidence that supports the claim.  While the empirical evaluation is restricted to discrete inference problems it represents an important contribution that will help practitioners decompose large scale problems into approximate local problems.